# Statistical characterization of erosion and sediment transport mechanics in shallow tidal environments. Part 1: erosion dynamics

Andrea D'Alpaos[1,*], Davide Tognin[1,2,*], Laura Tommasini[1], Luigi D'Alpaos[2], Andrea Rinaldo[2,3], and Luca Carniello[2]

[1]Department of Geosciences, University of Padova, Padova, Italy
[2]Department of Civil, Environmental, and Architectural Engineering, University of Padova, Padova, Italy
[3]Laboratory of Ecohydrology ECHO/IEE/ENAC, Ècole Polytechnique Fèdèrale de Lausanne, Lausanne, Switzerland
[*]These authors contributed equally to this work.

**Correspondence:** Davide Tognin (davide.tognin@unipd.it)

**Abstract.** Reliable descriptions of erosion events are foundational to effective frameworks relevant to the fate of tidal landscape evolution. Besides the rhythmic, predictable action of tidal currents, erosion in shallow tidal environments is strongly influenced by the stochastic wave-induced bottom shear stress (BSS), mainly responsible for sediment resuspension on tidal flats. However, the absence of sufficiently long, measured time series of BSS prevents a direct analysis of the combined tide- and wave-driven erosion dynamics and its proper representation in long-term morphodynamic models. Here we test the hypothesis of describing erosion dynamics in shallow tidal environments as a Poisson process by analysing with the peak-over-threshold theory the BSS time series computed using a fully-coupled, bi-dimensional numerical model. We perform this analysis on the Venice Lagoon, Italy, taking advantage of the availability of several historical surveys in the last four centuries, which allow us to investigate the effects of morphological modifications on spatial and temporal erosion patterns. Our analysis suggests that erosion events on intertidal flats can effectively be modelled as a marked Poisson process in different morphological configurations, because interarrival times, durations and intensities of the over-threshold exceedances are always well described by exponentially distributed random variables. The resulting statistical characterization allows a straightforward computation of morphological indicators, such as the erosion work, and paves the way to a novel synthetic, yet reliable, approach for the long-term morphodynamic modelling of tidal environments.

## 1 Introduction

Together with tidal currents, wind waves are key drivers of the morphological evolution of shallow tidal landscapes (Green and Coco, 2014). The morphology of tidal flats and subtidal platforms is mainly controlled by wave-induced erosion and resuspension, together with the rate of relative sea level rise and sediment supply (Fagherazzi et al., 2006; D'Alpaos et al., 2012; Hu et al., 2017; Zhou et al., 2017; Belliard et al., 2019). Moreover, the action of wind waves is usually recognized as one of the main causes of the retreat of salt-marsh margins (Möller et al., 1999; Marani et al., 2011; Bendoni et al., 2016; Leonardi et al., 2016; Finotello et al., 2020). Therefore, the temporal and spatial evolution of wave-induced bottom shear stresses (BSSs) has an important impact on sediment dynamics in the intertidal zone (Carniello et al., 2005; Fagherazzi and

Wiberg, 2009; Mariotti et al., 2010), ultimately influencing the morphological and biological processes responsible for the evolution of tidal systems. For instance, local wave-induced BSS can influence sediment winnowing and distribution on tidal flats (Zhou et al., 2015; Ghinassi et al., 2019; Zhou et al., 2022). Moreover, large wave-induced BSSs can disrupt the polymeric biofilm built up by microphytobenthos (MPB) typically colonizing the bed sediment in shallow tidal environments (Amos et al., 2004; Chen et al., 2017; Pivato et al., 2019) and therefore promote erosion of tidal-flat surfaces. However, the related increase in suspended sediment concentration, typically occurring during storm surges, can support sedimentation on salt marshes, thus helping them to keep pace with sea level rise (Goodbred and Hine, 1995; Tognin et al., 2021). Therefore, understanding BSS dynamics is fundamental to describe the morphodynamic evolution of shallow tidal systems and to indicate long-term sustainable management strategies (Zhou et al., 2022), which is crucial under the increasing pressure of relative sea level rise (Nicholls et al., 2021) and human interventions (Tognin et al., 2022).

Several process-based models have been developed to describe erosive processes and investigate the effects of BSS on the morphodynamics of shallow tidal basins. Although these models were originally developed to deal with short-term time scales, various techniques were proposed to accelerate bed evolution and upscale the results at much longer time scales. The most commonly adopted upscaling techniques are the tide-averaging procedure and the online update with a morphological factor (Latteux, 1995; Roelvink, 2006). The tide-averaging approach assumes that the time scale at which morphological changes take place is far longer than that typical of hydrodynamic changes and, therefore, the bottom can reasonably be considered fixed over a tidal cycle, so that bed level changes can be offline upscaled from gradients in the tidally-averaged sediment transport (e.g., Lanzoni and Seminara, 2002; Sgarabotto et al., 2021). On the contrary, to continuously update the morphology describing short-term interactions between hydrodynamics and sediment transport (e.g., Defina, 2003; Lesser et al., 2004; Carniello et al., 2012) but, at the same time, to overcome the prohibitively time-consuming computation associated with the direct upscaling of this fully-coupled approach, the so-called 'morphological factor' can be adopted (Lesser et al., 2004; Roelvink, 2006). The morphological factor $n$ is essentially an online multiplication factor for the bed change, so that the results of a simulation over one tidal cycle in fact depict the morphological changes over $n$ cycles. Fully-coupled models upscaled using a morphological factor are increasingly adopted to investigate not only decadal (e.g., Gourgue et al., 2022) but also centennial to millenial morphological evolution of tidal and estuarine systems (e.g., Braat et al., 2017; Pinton et al., 2023; Baar et al., 2023)

These upscaling techniques are based on the underlying assumption that the actual morphological evolution of a system is equivalent to that resulting from a repetitive pattern representative of the dominant forcing conditions. This hypothesis is reasonable when the main hydrodynamic forcing is represented by tidal oscillation, although attention should also be paid when selecting representative boundary conditions for astronomic tidal patterns (Schrijvershof et al., 2023). Instead, taking into account also merely stochastic processes, such as wind waves and storm surges, is far less straightforward. When wind climate may be reduced to a limited set of representative conditions, multiple simulations can be run and then a weighted average of the different results can be determined to estimate the upscaled morphological evolution (Roelvink, 2006). However, when representative wind and storm climate cannot be reduced to a limited batch of boundary conditions or, more importantly, not only the magnitude but also the temporal succession of these events is likely to strongly affect the morphological evolution of the system, these upscaling techniques cannot be properly applied.

A different perspective would be to directly consider the stochasticity of erosion dynamics. From this point of view, the first step is to test the possibility of effectively describing BSS dynamics within a statistically-based framework. Once verified, this hypothesis will allow one to generate synthetic, yet reliable, BSS time series to model the erosion dynamics on long-term time scales and compare the possible modifications also considering different scenarios in a computationally-effective way through the use of independent Monte Carlo realizations. Although the statistical characterization of the long-term behaviour of several geophysical processes is becoming increasingly popular in hydrology and geomorphology (e.g., Rodriguez-Iturbe et al., 1987; D'Odorico and Fagherazzi, 2003; Botter et al., 2007; Park et al., 2014), applications to tidal landscapes are still quite rare (D'Alpaos et al., 2013; Carniello et al., 2016).

The statistical characterization of erosion events on which this approach is based requires a sufficiently long BSS time series. To this purpose, direct, continuous BSS measurements are rarely available and, thus, process-based numerical models can represent a useful tool to provide sufficiently-long and spatially-distributed time series to characterize erosion dynamics at the basin scale. Moreover, to test the hypothesis that such a simplified approach holds independently on the specific morphological configuration, this characterization should ideally be performed in different tidal systems or in a single tidal system where information on the morphological evolution is available for a sufficiently long time. From this point of view, the Venice Lagoon, Italy (Figure 1) represents an almost unique opportunity to test long-term models, because several bathymetric surveys are available for the last four centuries (Carniello et al., 2009; D'Alpaos, 2010a; Tommasini et al., 2019; Finotello et al., 2023).

Towards the goal of developing a synthetic theoretical framework to represent wind-wave-induced erosion events and accounting for their influence on the long-term morphodynamic evolution of tidal systems, we applied a two-dimensional finite element model for reproducing and analysing the combined effect of wind waves and tidal currents in generating BSSs in several historical configurations of the Venice Lagoon. More in detail, in the present study, we used the fully coupled Wind Wave-Tidal Model (WWTM) (Carniello et al., 2005, 2011) to investigate the hydrodynamic behaviour of six historical configurations of the Venice Lagoon, namely 1611, 1810, 1901, 1932, 1970, and 2012. For each configuration, we run a one-year-long simulation considering representative tidal and meteorological boundary conditions. The resulting spatial and temporal dynamics of BSSs for the six selected configurations were analyzed on the basis of the peak-over-threshold (POT) theory once a critical shear stress for bed sediment erosion was chosen.

The main goal of this analysis is to find whether, in line with previous results on the present-day configuration of the Venice Lagoon (D'Alpaos et al., 2013), wave-induced erosion events can be modelled as marked Poisson processes also in different morphological settings. The relevance of this result lies in the possibility of describing erosion processes as a Poisson process over time, which represents a promising framework for long-term studies. Our analyses provide a temporally and spatially explicit characterization of wind-induced erosion events for the Venice Lagoon starting from the beginning of the seventeenth century, thus allowing us to investigate and understand the main features of the erosive trends the lagoon has been experiencing and to provide predictions on future scenarios.

## 2 Materials and Methods

### 2.1 Geomorphological setting

The Venice Lagoon, located in the northern Adriatic Sea and characterized by an area of 550 km$^2$, formed over the last 7500 years covering alluvial Late Pleistocene, silty-clayey deposits, locally known as Caranto (Zecchin et al., 2008). In the present-day morphology, the lagoon is connected to the sea with three inlets, namely Lido, Malamocco, and Chioggia (Figure 1), through which the tide propagates within the back-barrier system. The tidal regime is semidiurnal with a maximum tidal amplitude of about 0.75 m, typical of the northern Adriatic Sea (D'Alpaos et al., 2013; Valle-Levinson et al., 2021).

Meteorological conditions also importantly affect the hydrodynamics of the Venice Lagoon. In particular, storm surges generated by the south-easterly Sirocco wind (Figure 1b) often overlap astronomical tides, thus importantly increasing water levels (Mel et al., 2014). Whereas, the north-easterly Bora wind (Figure 1b) is mainly responsible for the generation of wind waves and water level set-up, especially in the central and southern portions of the lagoon (Carniello et al., 2009).

The morphology of the Lagoon deeply changed over the last four centuries (Figure 2), especially owing to anthropogenic modifications (Carniello et al., 2009; D'Alpaos, 2010a; Silvestri et al., 2018; Finotello et al., 2023). By the end of the 16$^{th}$ century, all the major rivers flowing into the lagoon were diverted to debouch directly into the open sea, thus dramatically decreasing fluvial sediment input. Furthermore, between the 1900s and 1950s, the exploitation of groundwater for industrial purposes accelerated the local subsidence rates, with anthropogenically-induced subsidence reaching values of about 10 to 14 cm in the area of the city of Venice (Carbognin et al., 2004; Zanchettin et al., 2021). In the same period, the total area open to the propagation of tides was largely reduced due to extensive land reclamation carried out to accommodate industrial, agricultural, and aquacultural activities, especially along the landward margin of the lagoon (Figure 2c-e). On the seaward side, massive jetties were built between 1839 and 1934 to stabilize the sections of the three inlets and maintain water depths requested for increasingly bigger commercial ships (Figure 2c,d). For the same reason, navigation channels were excavated in the central part of the lagoon to connect the inner harbour with the sea (Figure 2e). As a result, these human interventions, together with eustatic sea-level rise (average value 1.23 ± 0.13 mm/year between 1872 and 2019; 2.76 ± 1.75 mm/year between 1993 and 2019; see Zanchettin et al., 2021), played a primary role in affecting the morphological evolution of the lagoon. Only in the northern lagoon, the morphological degradation was less pronounced because of the sheltering effect provided by the mainland against the north-easterly Bora wind and the less intense human pressure. Therefore, the northern basin displays also in the present-day configuration relatively shallow intertidal flats and larger salt-marsh areas, compared to the central and southern lagoon (Figure 2f).

We considered six different configurations of the Venice Lagoon (Figure 2 and S1), covering a time span of four centuries, in order to assess the evolution through time of the feedback mechanisms between morphology and wave-induced erosion. The oldest three configurations (1611, 1810, and 1901) were reconstructed by using historical maps, while the more recent ones make use of the topographic surveys carried out by the Venice Water Authority (Magistrato alle Acque di Venezia) in 1932, 1970, and 2003. The updated description of the more recent morphological modifications, which mainly occurred at the three

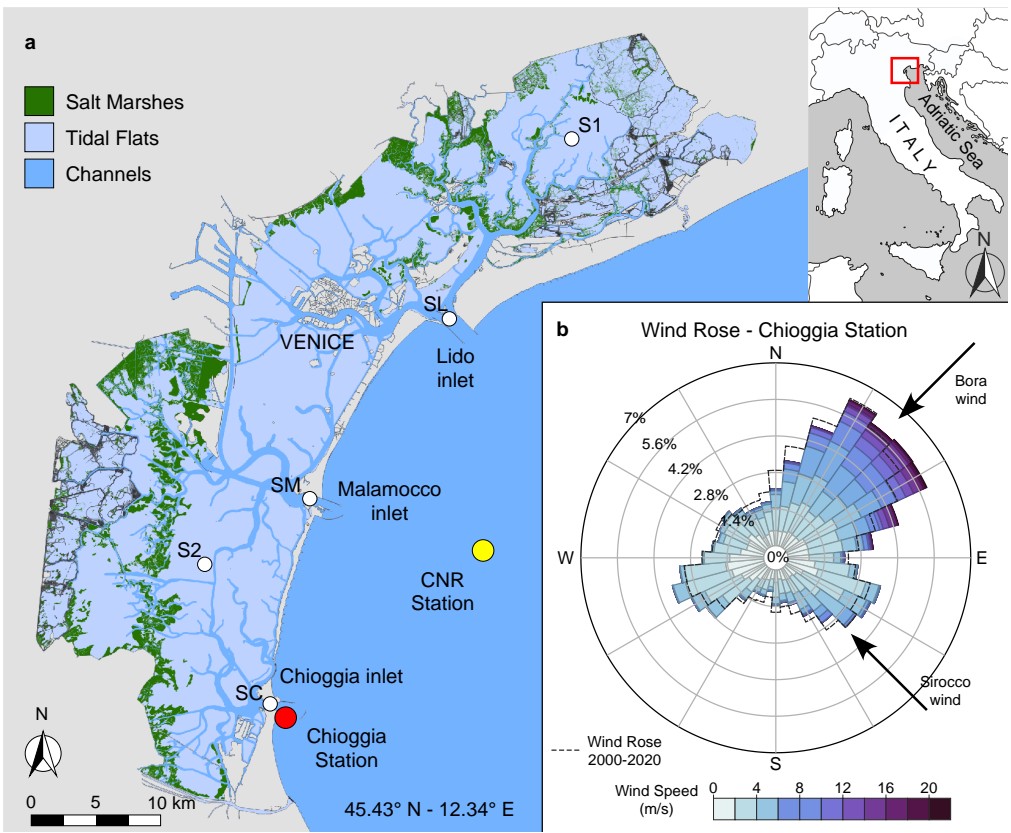

**Figure 1. Morphological features and wind conditions characterizing the Venice Lagoon. a**, Spatial distribution of the morphological features characterizing the Venice Lagoon. The locations of the anemometric (Chioggia) and oceanographic (CNR Oceanographic Platform) stations are also shown, together with the locations of the three stations at the inlets (SL, SM and SC) and two stations (S1 and S2) for which we provide a detailed statistical characterization of over-threshold events. **b**, Wind rose for the data recorded at the Chioggia station in 2005. The dashed line shows the wind rose for the period 2000-2020.

inlets in the context of the Mo.S.E. project for the safeguard of the city of Venice by high tides (almost completed in 2012), was included in the 2003 configuration, so that we refer to the latter configuration as to the 2012 configuration.

In each bathymetry and, hence, in each computational grid, bed elevation refers to the local mean sea level at the time when each survey was performed. We refer the reader to Tommasini et al. (2019) for a detailed description of the methodology adopted to reconstruct the historical configurations and for information on the bathymetric data of the Venice Lagoon. The computational grids reproducing all the six considered configurations of the Lagoon are shown in Figure S1 and were calibrated in previous studies, namely: 1611 by Tommasini et al. (2019); 1810 by D'Alpaos and Martini (2005) and D'Alpaos (2010b);

1901, 1932, 1970 and 2012 by Carniello et al. (2009) and Finotello et al. (2023).

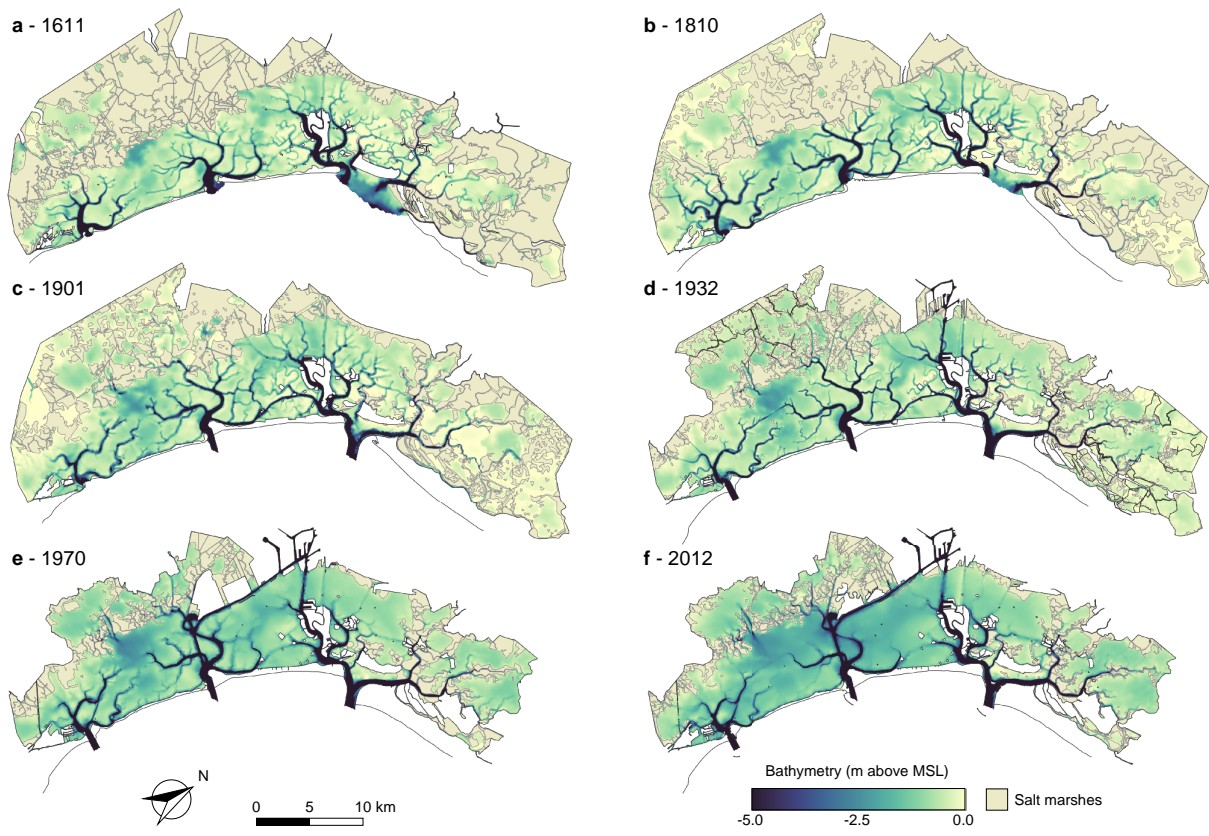

**Figure 2. Historical bathymetries of the Venice Lagoon.** Color-coded bathymetries of the six different configurations of the Venice Lagoon: (**a**) 1611, (**b**) 1810, (**c**) 1901, (**d**) 1932, (**e**) 1970, and (**f**) 2012.

## 2.2 Numerical Model and Simulations

To compute the hydrodynamic and the wind-wave fields in the six selected configurations of the Venice Lagoon, we used the two-dimensional (2-D) fully coupled Wind Wave-Tidal Model (WWTM) (Carniello et al., 2005, 2011). The numerical model, coupling a hydrodynamic module and a wind-wave module, describes the hydrodynamic flow field together with the generation and propagation of wind waves using the same computational grid.

The hydrodynamic module uses a semi-implicit staggered finite element method based on Galerkin's approach to solving the 2-D shallow water equations suitably rewritten in order to deal with partially wet and morphologically irregular domains (Defina, 2000; Martini et al., 2004, Supplementary information). The bottom shear stress induced by currents, $\tau_{tc}$, is evaluated using the Strickler equation considering the case of a turbulent flow over a rough wall, which can be written as (Defina, 2000)

$$\tau_{tc} = \rho g Y \left( \frac{|\boldsymbol{q}|}{K_s^2 H^{10/3}} \right) \boldsymbol{q} \tag{1}$$

where $\rho$ is water density, $g$ is the gravity acceleration, $Y$ is the effective water depth, defined as the volume of water per unit area actually ponding the bottom, $\boldsymbol{q}$ is the flow rate per unit width, $K_s$ is the Strickler roughness coefficient, and $H$ is an equivalent water depth that accounts for the typical height of ground irregularities. The hydrodynamic module also provides the water levels that are used by the wind-wave module to assess the wave group celerity and the bottom influence on wind-wave propagation.

For the wind-wave module (Carniello et al., 2005, 2011), the wave action conservation equation is parameterized using the zero-order moment of the wave action spectrum in the frequency domain (Holthuijsen et al., 1989, Supplementary information). An empirical correlation function relating the peak wave period to the local wind speed and water depth determines the spatial and temporal distribution of the wave period (Young and Verhagen, 1996; Breugem and Holthuijsen, 2007; Carniello et al., 2011). The wind-wave module computes the bottom shear stress induced by wind waves as (Carniello et al., 2005)

$$\tau_{ww} = \frac{1}{2}\rho f_w u_m^2 \tag{2}$$

where $u_m$ is the maximum horizontal orbital velocity associated with wave propagation and $f_w$ is the wave friction factor. According to the linear theory, the bottom velocity $u_m$ can be evaluated as

$$u_m = \frac{\pi H_w}{T \sinh(kh)} \tag{3}$$

where $H_w$ is the wave height, $T$ denotes the wave period, $k$ is the wave number, and $h$ is the water depth. The wave friction factor can be approximated as (Soulsby, 1997)

$$f_w = 1.39 \left[ \frac{u_m T}{2\pi(D_{50}/10)} \right]^{-0.52} \tag{4}$$

where $D_{50}$ is the median grain size.

The total bottom shear stress, $\tau_{wc}$, resulting from the combined effect of tidal currents and wind waves, is enhanced beyond the sum of the two contributions, because of the non-linear interaction between the wave and the current boundary layer. In the WWTM this is accounted for by using the empirical formulation suggested by Soulsby (1995, 1997):

$$\tau_{wc} = \tau_{tc} + \tau_{ww} \left[ 1 + 1.2 \left( \frac{\tau_{ww}}{\tau_{ww} + \tau_{tc}} \right) \right] \tag{5}$$

Even if BSSs induced by the tidal currents are typically smaller than those produced by wind waves, they are of fundamental importance in modulating the temporal evolution of the total BSSs and can increase the peak BSS values by up to 30% (Mariotti et al., 2010; D'Alpaos et al., 2013).

The WWTM has been widely tested against field observations not only in the Venice Lagoon (e.g., Carniello et al., 2005, 2011; Tognin et al., 2022) but also in other shallow microtidal environments worldwide, for example in the back-barrier lagoons of the Virginia Coast Reserve (Mariotti et al., 2010) and the Cádiz Bay (Zarzuelo et al., 2018, 2019). Concerning the Venice Lagoon, model calibration and testing have been performed only in the most recent configuration, i.e., when field data are available. For the older configurations of the lagoon, no hydrodynamic measurements are available and, consequently, the roughness coefficient values have been derived in analogy from those selected for the calibrated grid in the most recent configuration, by

comparing local sediment grain size, bed elevation and morphological classes (e.g. channel, tidal flat, salt marsh), which also take into account for the possible presence of vegetation (Finotello et al., 2023).

We summarise here the model performance in reproducing tidal levels, significant wave heights and flow rates at the inlets by reporting the standard Nash-Sutcliffe Model Efficiency (NSE) parameter computed when field data are available and refer the interested reader to the Supplementary Information (Figures S2 to S5) and the literature (Carniello et al., 2005, 2011; Tognin et al., 2022) for a more detailed comparison. Adopting the classification proposed by Allen et al. (2007), the model performance can be rated from excellent to poor (i.e., NSE > 0.65 excellent; 0.5 < NSE < 0.65 very good; 0.2 < NSE < 0.5 good; NSE < 0.2 poor). The WWTM model is excellent in reproducing tidal levels ($NSE_{mean}$ = 0.970, $NSE_{median}$ = 0.984, $NSE_{std}$ = 0.040), very good to excellent in reproducing significant wave heights ($NSE_{mean}$ = 0.627, $NSE_{median}$ = 0.756, $NSE_{std}$ = 0.357), and excellent in replicating flow rates at the inlets ($NSE_{mean}$ = 0.853, $NSE_{median}$ = 0.931, $NSE_{std}$ = 0.184) (Statistics are derived from calibration reported in Carniello et al. (2011), their Tables 1,2, and 3, and Tognin et al. (2022), their Table S2).

We applied the numerical model to the six computational domains representing the Venice Lagoon and a portion of the Adriatic Sea in front of it in order to perform one-year-long simulations (Figure S1). The boundary conditions of the model are the hourly tidal levels measured at the Consiglio Nazionale delle Ricerche (CNR) Oceanographic Platform, located in the Adriatic Sea approximately 15 km offshore of the coastline, and wind velocities and directions recorded at the Chioggia anemometric station, for which a quite long data set was available (Figure 1a). In particular, we selected as boundary conditions measurements recorded in 2005, because they can be considered representative of the wind climate of the Venice Lagoon. Indeed, the probability distribution of wind speeds in 2005 shows the minimum difference with the mean probability distribution computed for the period 2000-2020, compared to any other year within the same time interval (Supplementary information, Figure S6 and Table S1). Indeed, a visual comparison between the wind roses for 2005 and the entire period 2000-2020 supports this choice (Figure 1b). Because bed elevation in each computational grid refers to the coeval mean sea level, by using the same water levels as boundary conditions we implicitly take into account the effect of historical relative sea level variations. Considering the same wind and tidal forcing in each historical configuration of the Venice Lagoon allowed us to isolate the effects of the morphology on the hydrodynamics and wind-wave fields.

## 2.3 Peak Over Threshold Analysis of BSS

The morphodynamic evolution of tidal environments is controlled by the complex interaction among hydrodynamic, biologic and geomorphologic processes, which include both deterministic and stochastic components. As an example, it was shown that sediment transport dynamics in the Venice Lagoon is mostly linked to some limited and severe events induced by wind waves (Carniello et al., 2011), whose dynamics are markedly stochastic in the present-day configuration of the lagoon (D'Alpaos et al., 2013; Carniello et al., 2016). In this work, at any location within each considered configuration of the Venice Lagoon, we used the peak-over-threshold (POT) theory (Balkema and de Haan, 1974) to analyze the temporal and spatial evolution of the total BSS, $\tau_{wc}$.

In general, the selection of the threshold for the POT method must satisfy two contrasting requirements. On the one hand, the threshold must be large enough to discern stochastic events from the deterministic background. On the other hand, the threshold should not be too high to avoid the loss of important information and the need for a much longer time series to compute meaningful statistics, because of the lower number of threshold exceedances. Moreover, the extreme value theory postulates the general emergence of Poisson processes whenever the censoring threshold is high enough (Cramér and Leadbetter, 1967). To comply with these requirements, in the present analysis, the threshold is maintained well below the maximum observed values, in order to remove only the background modulation induced by tidal currents without losing significant information on the stochastic wave-driven erosion process.

In applying the POT method to BSS time series, setting the threshold equal to a critical BSS for erosion, $\tau_c$, presents the non-trivial advantage of preserving also the physical meaning of the erosion mechanism. Values of critical BSS for erosion for fine, cohesive mixtures typical of shallow tidal settings largely vary in the literature and are affected by multiple physical and biotic factors (Mehta et al., 1989). Erosion shear stress from in-situ measurements on the tidal flats of the Venice Lagoon ranges between 0.2 and 2.3 Pa ($0.7 \pm 0.5$ Pa - median $\pm$ standard deviation), with values higher than 0.9 Pa usually recorded within densely vegetated patches (Amos et al., 2004). In the present analysis, we cannot take into account the role of the biotic component, because of the impossibility to reconstruct the spatial distribution of vegetated tidal flats in the ancient configurations of the Venice Lagoon. For all the above reasons and following the approach suggested by D'Alpaos et al. (2013), we set the critical shear stress, $\tau_c$, equal to 0.4 Pa for all the selected historical configurations of the Venice Lagoon.

Before performing the POT analysis, the time series of BSSs were processed by applying a moving average filter, in order to remove spurious upcrossings and downcrossings of the prescribed threshold. This low-pass filter with a time window of 6 hours removes short-term fluctuations, preserving the modulation given by the semidiurnal tidal oscillation. Thanks to this preprocessing procedure, over-threshold events satisfy the independence assumption required by the statistical analysis applied.

The POT method allowed us to identify:

1. the interarrival time of over-threshold events, defined as the time between two consecutive upcrossings of the threshold;

2. the duration of over-threshold events, that is the time elapsed between any upcrossing and the subsequent downcrossing of the threshold;

3. its intensity, calculated as the largest exceedance of the threshold in the time elapsed between an upcrossing and the following downcrossing.

These three random variables synthetically characterize the over-threshold erosion events and can be combined to obtain further metrics to describe the erosion process (e.g. see the erosion work defined later on).

Once the probability density functions and the corresponding moments of these variables were defined, a statistical analysis was performed for each location in all the considered configurations of the Venice Lagoon, in order to provide an accurate description of the BSS evolution through the last four centuries. This enabled us to highlight the feedback between morphology and resuspension events over long-term time scales.

We performed the non-parametric Kolmogorov-Smirnov (KS) goodness of fit test to verify the hypothesis that the inter-arrival time of over-threshold events is an exponentially distributed random variable. The interarrival probability distribution plays an important role because, if interarrival times between subsequent exceedances of the threshold $\tau_c$ are independent and exponentially distributed random variables, the mechanics of erosion events can be mathematically described as a 1-D marked Poisson process, characterized by a vector of random marks (intensity and duration of each over-threshold event) associated to a sequence of random events along the time axis (Cramér and Leadbetter, 1967; Gallager, 2013). Memorylessness is one of the most interesting mathematical features of Poisson processes since it allows one to set the probability of observing a certain number of events in a pre-established time interval dependent only on its duration, regardless of its position along the time series (Gallager, 2013). Therefore, the description of over-threshold BSS events as a Poisson process will allow one to immediately identify the probabilities of observing a certain number of resuspension events in a year or during a season, because all the sources of stochasticity in the physical drivers are described by a single parameter (i.e. the mean frequency of the process). This suggests the possibility of setting up a synthetic theoretical framework to model the wave-induced events through the use of Monte-Carlo realizations, bearing important consequences for the long-term evolution of tidal landscapes.

The result of modelling erosion events as a Poisson process stands regardless of the specific value of the censoring threshold selected for the POT analysis, provided that it is high enough to exclude deterministic exceedances, and this is confirmed also by the sensitivity analysis performed by D'Alpaos et al. (2013) on the present-day configuration of the Venice Lagoon. Indeed, when considering too low values of the threshold (e.g. $\tau_c = 0.2$ Pa), deterministic exceedances driven by tidal currents occur and make the interarrival time not exponentially distributed. On the contrary, as the threshold value increases (e.g. $\tau_c \geq 0.6$ Pa), the KS test is still verified, thus confirming that the process remains Poisson for increasing censoring thresholds (see Figure 6 in D'Alpaos et al. (2013) for further details).

## 3   Results and Discussion

We analyzed the time series of computed total BSSs, $\tau_{wc}$, at any element of the computational grids reproducing the six selected configurations of the Venice Lagoon on the basis of the POT method, in order to characterize the over-threshold erosion events in terms of interarrival times, peak excess and duration. The KS test is then performed in each element of the six domains in order to test where interarrival times can be described by an exponential distribution and thus, the over-threshold erosion events can be modelled as a Poisson process. We performed the KS test also on peak excess and duration to test if these marks of the process can also be described by exponential distributions. Whether peak excess and duration can be described by exponential distributions does not affect the chance to model erosion as a Poisson process, which indeed relies only on the exponentiality of interarrival times, but it can simplify the setup of the final stochastic framework. Therefore, in the spatial distribution of the KS test results (Figure 3), we distinguished:

1. the dark blue area, where the KS test is not verified for the interarrival time, i.e. wave-induced erosion events can not be described as a Poisson process;

2. the red area, where the KS test is verified for all the three considered stochastic variables, namely interarrival times, intensity, and duration, i.e. wave-induced erosion events are indeed a marked Poisson process where its marks, intensity and duration, are also exponentially distributed random variables;

3. the yellow area, where the KS test is verified for the interarrival time but it is not verified for the intensity and/or duration, i.e. wave-induced erosion events are a marked Poisson process but at least one between intensity and duration is not an exponentially distributed random variable.

The mean interarrival times (Figure 4), mean peak excesses (Figure 5) and mean durations of over-threshold erosion events (Figure 6) in the six selected configurations of the Venice Lagoon are shown in every location where the KS test is satisfied for interarrival times (Figure 3), and, thus, erosion events, can be described as a Poisson process. Mean peak excess and mean duration are shown also where at least interarrival times are exponentially distributed (i.e. yellow areas in Figure 3) because mean values are anyway considered to be informative and erosion events can still be modelled as a Poisson process, although the marks are described by a distribution different from the exponential one.

Wind-wave generation is determined by energy transfer from the wind to the water surface and, thus, it clearly depends on wind characteristics, namely wind intensity and duration, as well as on fetch length and water depth (Fagherazzi et al., 2006; Fagherazzi and Wiberg, 2009). As a consequence, the spatial distribution and morphological characteristics of channels, tidal flats, and, more importantly, salt marshes and islands strongly influence the response of a shallow tidal basin to wind forcing and the resulting distribution of BSSs (Fagherazzi et al., 2006; Defina et al., 2007). Large portions in the ancient configurations of the lagoon were occupied by salt-marsh areas, continuously interrupting the fetch and thus reducing the exceedances of the critical threshold. As a result, in the four more ancient configurations the characteristics of erosion events globally display a more complex spatial pattern, which conversely tends to be more uniform in the recent most configurations, due to the reduction in salt-marsh areas, to the increase in fetch length, and to the deepening of tidal flats.

In all the selected configurations, salt marshes and tidal channel networks mostly represent the portion of the lagoon where wave-induced erosion events cannot be modelled as a Poisson process (dark blue area in Figure 3). Over salt-marsh platforms almost no exceedances of the prescribed threshold, $\tau_c$, tend to occur (Figure S7) because of the low water depth that prevents the formation of significant waves (e.g., Möller et al., 1999). May we add that colonization of the salt-marsh surface by halophytic vegetation almost completely prevents any vertical erosion (Christiansen et al., 2000; Temmerman et al., 2005). On the contrary, exceedances of the threshold can be detected along the channel network and at the three inlets (Figure S7), but these are mostly associated with shear stresses produced by tidal currents, especially after the construction of the jetties at the inlets. Consequently, at these points the KS test is not satisfied and erosion events cannot be modelled as a Poisson process because of the strictly deterministic nature of tide-induced shear stress.

Particularly interesting is the temporal evolution of BSS at the inlets. For instance, the time series and the interarrival time probability distribution of over-threshold erosion events at the SM station in the Malamocco inlet (see Figure 1a for the location) clearly show how the morphological modifications affected the BSS (Figure 7). In the 1611 and 1810 simulations, in the absence of jetties at the inlets, the BSS was very small, so that the number of exceedances of the threshold was too low to be

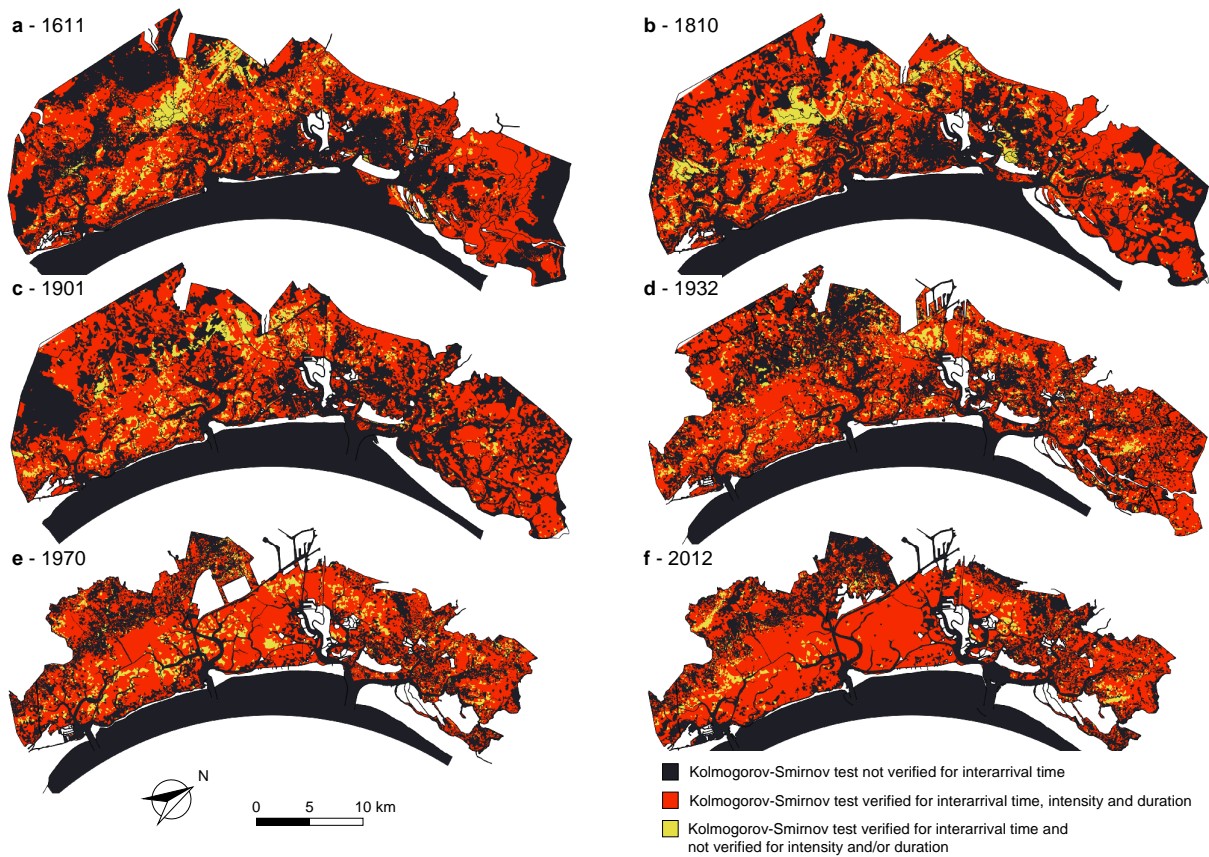

**Figure 3. Kolmogorov-Smirnov test for over-threshold erosion events.** Spatial distribution of Kolmogorov-Smirnov (KS) test at significance level ($\alpha = 0.05$) for the six different configurations of the Venice Lagoon: (**a**) 1611, (**b**) 1810, (**c**) 1901, (**d**) 1932, (**e**) 1970, and (**f**) 2012. In the maps we can distinguish areas where the KS test is: not verified (dark blue); verified for all the considered stochastic variables (interarrival time, intensity over the threshold and duration) (red); verified for the interarrival time and not for intensity and/or duration (yellow).

representative (Figure 7a,b). After the construction of the jetties at the Malamocco inlet in 1872, erosion mechanics abruptly changed: BSS considerably increased but it was driven by tidal forcing and, thus, interarrival times were not exponentially distributed, since the erosion threshold was exceeded on average once per day because of tidal fluxes (Figure 7c-f). The BSS analysis at the SL station in the Lido inlet, where the construction of the jetties ended in 1892, provides analogous results (Figure S8). Whereas, at the SC station in the Chioggia inlet, BSS still does not systematically exceed the threshold also in the 1901 configuration, since the construction of the jetties at the Chioggia inlet took place between 1930 and 1934 (D'Alpaos, 2010b) (Figure S9).

The KS test is verified over subtidal platforms and tidal flats, where current-induced BSSs are typically below the critical value, but wave-induced BSSs mainly contribute to the total BSS. Locations where interarrival time, duration and intensity

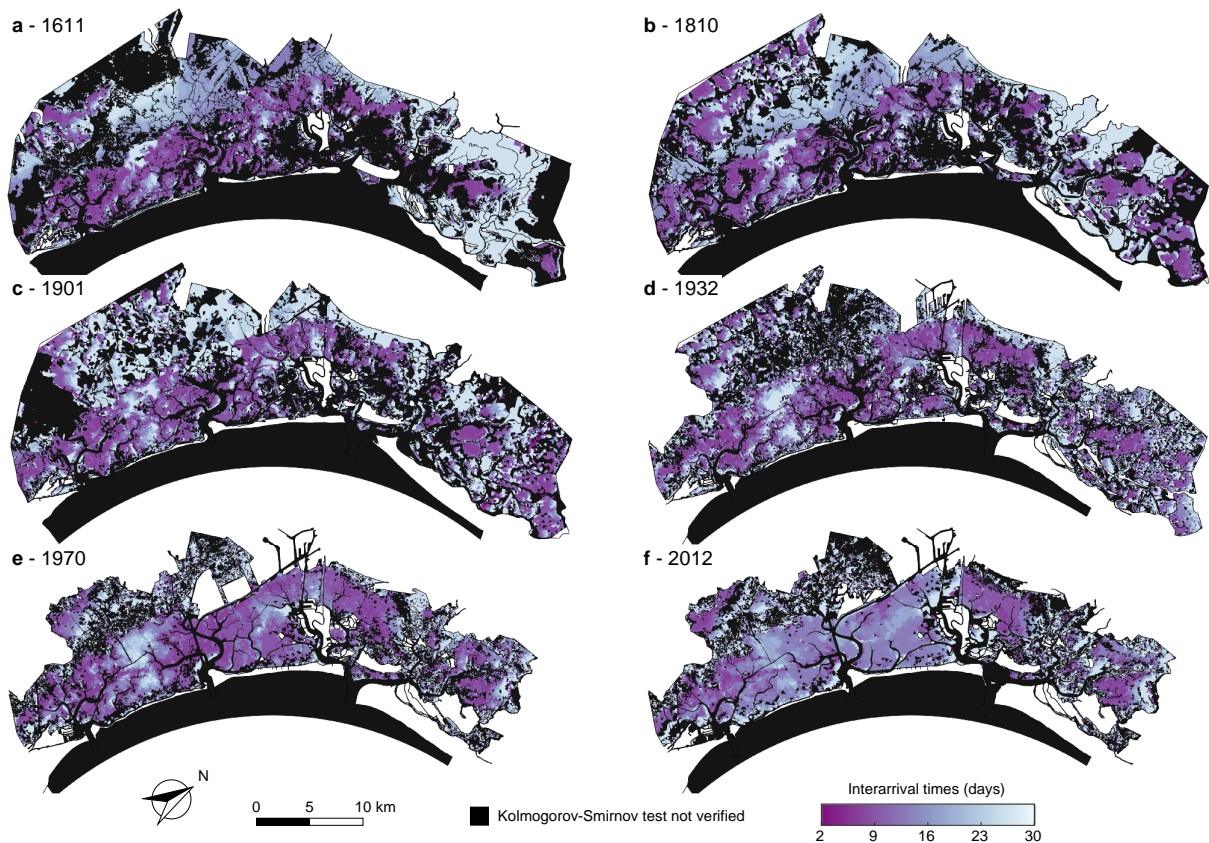

**Figure 4. Mean interarrival time of over-threshold erosion events.** Spatial distribution of mean interarrival times of over-threshold exceedances at sites where bed shear stress can be modelled as a marked Poisson process, as confirmed by the KS test ($\alpha = 0.05$) for the six different configurations of the Venice Lagoon: (**a**) 1611, (**b**) 1810, (**c**) 1901, (**d**) 1932, (**e**) 1970, and (**f**) 2012.

follow an exponential distribution (see red areas in Figure 3), remain the vast majority of the tidal basin in all the configurations. As a result, a synthetic framework that models erosion as a Poisson process is deemed to be suitable for wide tidal-flat areas. More generally, the chance to model erosion as a Poisson process lies in the intrinsic nature of BSS drivers. Wherever the stochastic action of wind waves and storm surges plays a prominent role in generating BSS compared to the deterministic tidal component, erosion is likely to be properly described by a Poisson process. This is the case of shallow tidal environments where the open water surface allows for the generation of wind waves, such as in back-barrier lagoons. On the contrary, the chance to use the Poisson-process-based approach diminishes where tidal currents substantially modulate BSS dynamics and mask the signature of stochastic processes, such as in tidal inlets and narrow meso- or macrotidal estuaries.

Almost in all configurations, large interarrival times (Figure 4) are essentially found in sheltered areas, where only particularly intense events are able to generate BSSs large enough to exceed $\tau_c$. A clear example is provided by the area protected by marsh platforms and by the mainland in the northeastern and in the western portion of the lagoon, sheltered from the north-

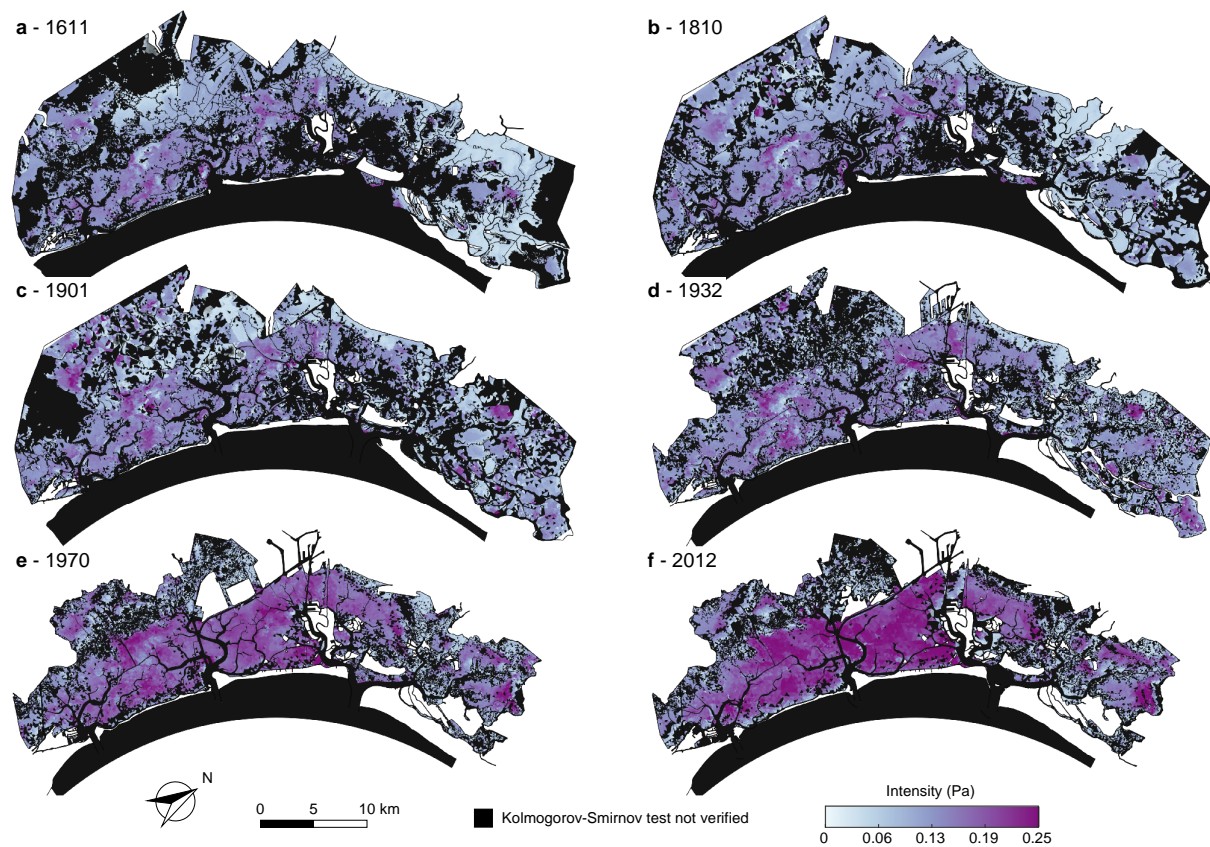

**Figure 5. Mean intensity of over-threshold erosion events.** Spatial distribution of mean intensity of peak excesses of over-threshold exceedances at sites where bed shear stress can be modelled as a marked Poisson process, as confirmed by the KS test ($\alpha = 0.05$) for the six different configurations of the Venice Lagoon: (**a**) 1611, (**b**) 1810, (**c**) 1901, (**d**) 1932, (**e**) 1970, and (**f**) 2012.

easterly Bora wind, which is the main morphologically significant wind in the Venice Lagoon (Figure 1b). This pattern becomes even more evident in the configurations of 1611, 1810, and 1901 where portions of the lagoon occupied by salt marshes are
wider than in the more recent configurations and display interarrival times longer than 30 days. Large interarrival times can also be observed close to the three inlets where the water depth is such that only during intense events the bottom can be affected by wave oscillations and the total BSSs can exceed the threshold. Globally speaking, in the four oldest configurations we found relatively short (about 5 days) interarrival times spread all around the lagoon, while the present configuration, characterized by a more uniform and larger water depth (in some areas deeper than 1.5 m), displays longer interarrival times, e.g. between 10
and 15 days for the tidal flats located in the central-southern portion of the lagoon (Figure 4 and S10a). This is mainly due to the relationship existing between $\tau_{ww}$ and water depth that, for a prescribed wind velocity, decreases as the water depth increases (Defina et al., 2007). Indeed, in the historical configurations large areas occupied by tidal flats are characterized by lower water

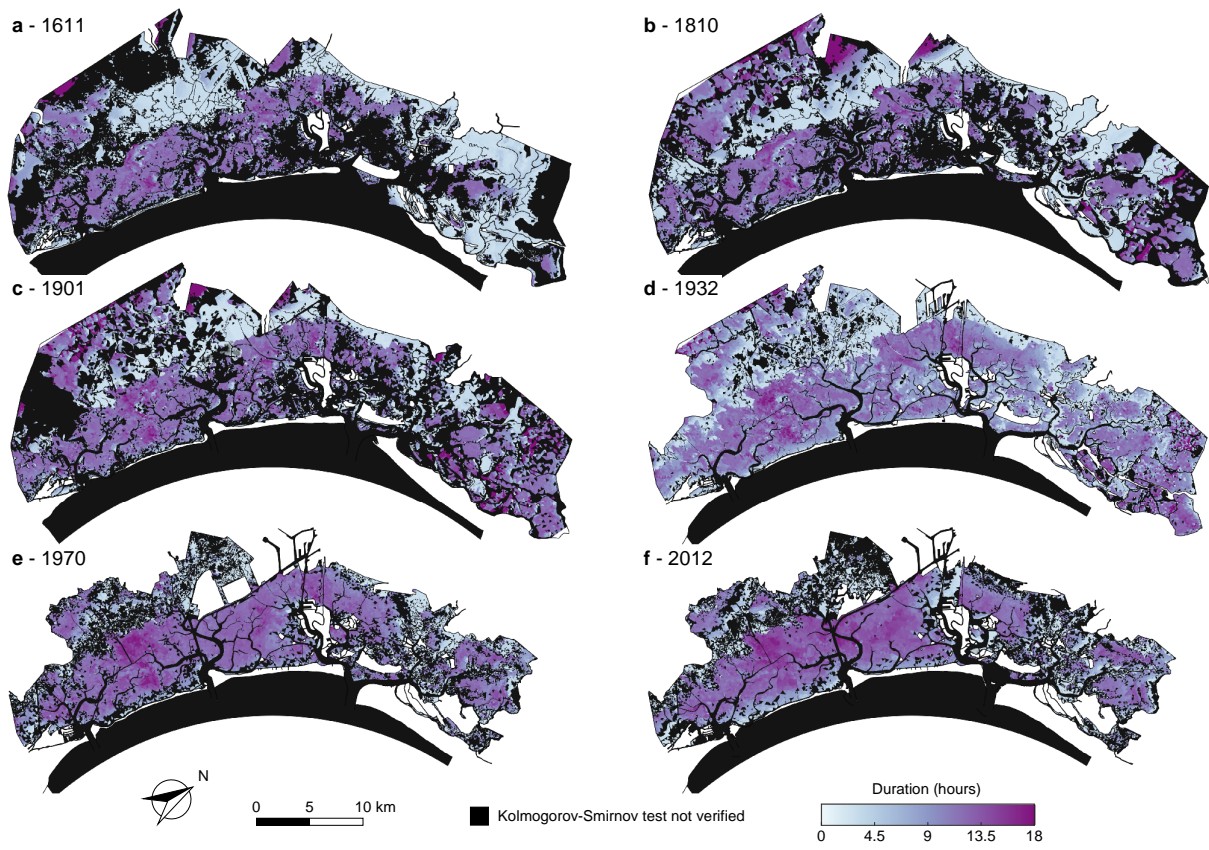

**Figure 6. Mean durations of over-threshold erosion events.** Spatial distribution of mean durations of over-threshold exceedances at sites where bed shear stress can be modelled as a marked Poisson process, as confirmed by the KS test ($\alpha = 0.05$) for the six different configurations of the Venice Lagoon: (**a**) 1611, (**b**) 1810, (**c**) 1901, (**d**) 1932, (**e**) 1970, and (**f**) 2012.

depth ($\leq 0.5$ m), and, as a result, $\tau_{ww}$ is higher also for weak wind speeds, thus increasing the number of exceedances of the threshold.

A punctual comparison among different configurations provide further insight into the effects of morphological changes on interarrival times (Figure 8). On a tidal flat in the northern lagoon named "Palude Maggiore" (see station S1 in Figure 1a), as in most areas of the lagoon, the mean interarrival time $\overline{\lambda}_t$ between two subsequent over-threshold events increases through time (Figure 8a). This is because this area preserved the same morphological features, i.e. relatively shallow tidal flats protected by the mainland and salt marshes, over the last four centuries. On the contrary, the tidal flat in the watershed divide area between

the Chioggia and the Malamocco inlets, named "Fondo dei Sette Morti" (see point S2 in Figure 1a), shows a reverse trend: the interarrival times decrease in time from 1611 to nowadays (i.e. wave-induced erosion events are more frequent, Figure 8d). Although the almost constant, relatively deep bottom elevation that characterized this area through centuries (Carniello et al., 2009; D'Alpaos, 2010b) prevents the exceedance of the threshold $\tau_c$ during less intense erosion events, the generalized deep-

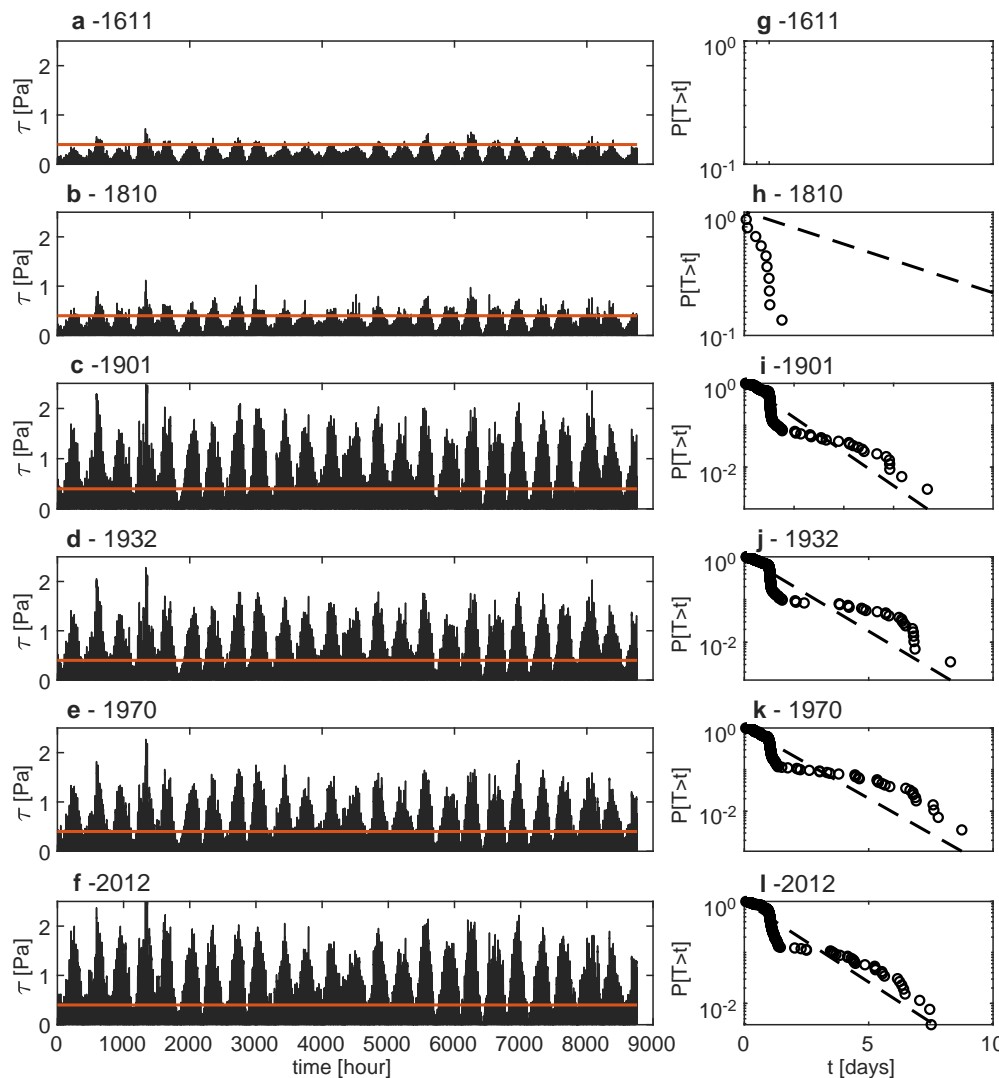

**Figure 7. Over-threshold BSS events at the Malamocco inlet.** Statistical analysis at SM station in the Malamocco inlet: (**a-f**) time series of the computed BSS; (**g-l**) probability distributions of the interarrival times (circles) and exponential distributions (dashed lines).

ening experienced by the surrounding portion of the lagoon in the most recent configurations promotes more frequent and less
intense events within this area and, therefore, a decrease of the interarrival times.

The over-threshold peak intensities generally strongly increased during the last four centuries (Figure 5 and S10b). For all the selected configurations, intensities are lower in the more pristine northern part of the lagoon, which is sheltered from

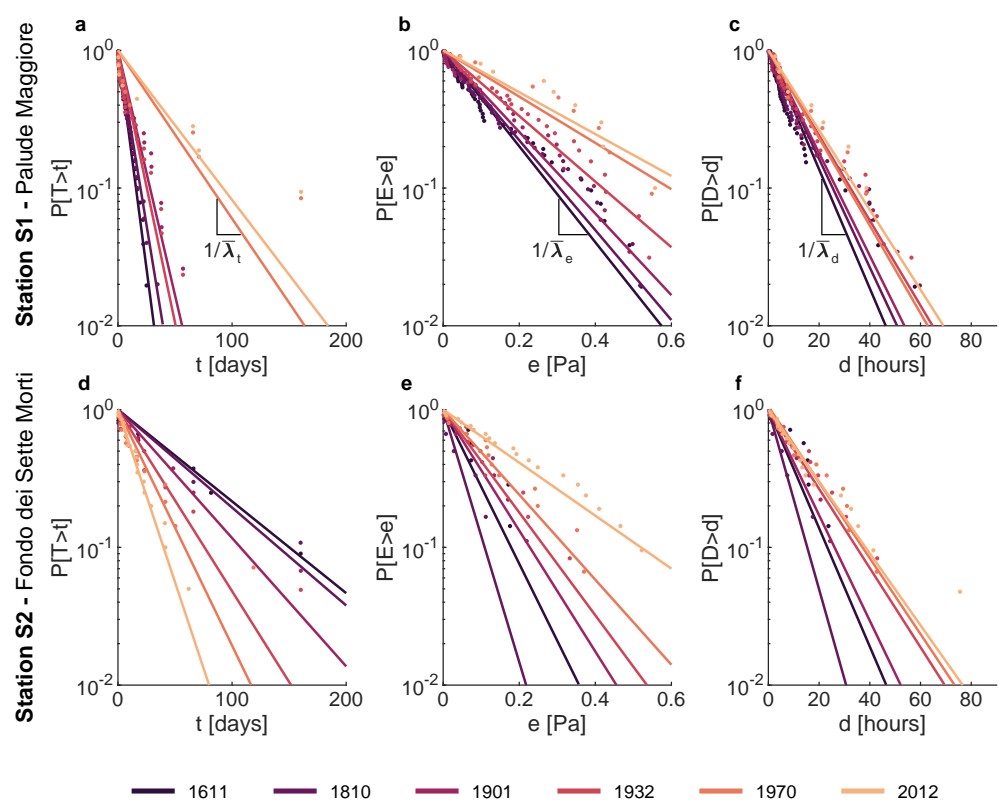

**Figure 8. Over-threshold erosion events at stations S1 and S2.** Statistical characterizations of over-threshold events at two stations S1 "Palude Maggiore" and S2 "Fondo dei Sette Morti" (see Figure 1a for locations) in the six configurations of the Venice Lagoon. Probability distributions of (**a-b**) interarrival times, $t$; (**c-d**) intensities of peak excesses of over-threshold exceedances, $e$; and (**e-f**) durations of over-threshold event, $d$. $\overline{\lambda}_t$ mean interarrival time, $\overline{\lambda}_e$ mean peak excess intensity, and $\overline{\lambda}_d$ mean duration.

the dominant Bora wind by the mainland and by preserved salt-marsh areas, interrupting the fetch. Whereas the central and southern portions of the lagoon are characterized by much larger intensity values, which more rapidly increased over the last
few decades. In particular, in the central part of the lagoon the mean intensities increased from around 0.13 Pa to 0.25 Pa above the threshold, due to the flattening and deepening of this area. A quite similar situation characterizes also the southern part of the Venice Lagoon, between the Malamocco and Chioggia inlets.

For all the configurations investigated, the durations of over-threshold events (Figure 6 and S10c), likewise intensities, present much lower values in the areas sheltered by salt marshes (i.e. the northern lagoon and the western portion of the
355 southern lagoon) than in the fetch-unlimited central-southern portion of the lagoon. In the latter area, indeed, over-threshold events last more than 15 hours, compared to a duration of about 5 hours in the more sheltered areas. The increase of peak intensities and durations of erosion events over time is also clearly shown by the probability distributions computed at points S1 and S2 (Figure 8).

The larger over-threshold peak intensities, as well as the longer durations characterizing the central-southern portion of the

lagoon and increasing from the past to the present, are in agreement with recent observations highlighting a critical erosive trend for the tidal flats and subtidal platforms in this area (Carniello et al., 2009; Molinaroli et al., 2009; D'Alpaos, 2010b; Defendi et al., 2010; Sarretta et al., 2010).

To investigate the relationship among the three random variables, the temporal cross-correlation is computed for each location and for all six configurations (Figures S11, S12 and S13). In particular, the temporal cross-correlation between intensity of

peak excesses and duration of over-threshold exceedances display values very close to 1 for all the lagoon morphologies, thus suggesting a pseudo-deterministic link between peak intensities and the corresponding durations (Figure S11 and S14a). On the contrary, almost no correlation exists between durations and interarrival times (Figure S12 and S14b), as well as between intensities and interarrival times (Figure S13 and S14c). These results, in line with the temporal cross-correlation obtained for the statistical analysis of suspended sediment concentration for the present lagoon by Carniello et al. (2016), suggest that resus-

pension events can be modelled as a 3-D Poisson process in which the marks (duration and intensity) are mutually dependent but independent from the interarrival time between two subsequent over-threshold events.

In order to provide a more quantitative estimation of the spatial heterogeneity of interarrival times, duration and intensities of the critical BSS exceedances, we computed the "erosion work" (geomorphic work *sensu* Wolman and Miller (1960), see also Mariotti and Fagherazzi (2013)), which represents the total amount of sediment resuspended during a selected time interval

and, thus, the potential erosion, because it does not consider any possible subsequent deposition. The erosion work $[E_w^*]$ experienced by a single point during the time interval $(t_2 - t_1)$ can be computed as:

$$[E_w^*] = \int_{t_1}^{t_2} \frac{e}{\rho_b} \left( \frac{\tau_{wc} - \tau_c}{\tau_c} \right) \mathrm{d}t. \tag{6}$$

where $e$ is the value of the erosion coefficient which depends on the sediment properties and $\rho_b = \rho_s(1-n)$ is the sediment bulk density, being $n$ the porosity. We set $\rho_s = 2650 \, \mathrm{kg \, m^{-3}}$ and $n = 0.4$ in agreement with Carniello et al. (2012). In principle, when computing the actual erosion, thus, taking into account both erosive and depositional processes, the parameter $e$ could

be calibrated by comparing the modelled erosion with that retrieved from the comparison among subsequent surveys provided that other non-natural processes (e.g., boat wave, dredging) do not strongly affect the local morphological evolution. However, because here we are considering the total potential rather than the actual erosion, such a calibration would not be correct. For this reason, we set $e$ equal to $5 \cdot 10^{-5} \, \mathrm{kg \, m^{-2} \, s^{-1}}$, as usually suggested for sand-mud mixtures (van Ledden et al., 2004; Le Hir et al., 2007).

Using the mean values of the stochastic variables considered herein (i.e. interarrival time, intensity and duration), once verified they can be modelled as a Poisson process, we can simplify Eq. 6 as follows:

$$[E_w] = \frac{e}{\rho_b} \left( \frac{\tau_{wc} - \tau_c}{\tau_c} \right) (t_2 - t_1) \tag{7}$$

where we assume $(t_2 - t_1)$ to be the mean duration of over-threshold events and $(\tau_{wc} - \tau_c)$ their mean intensity. In order to estimate the erosion work for one year, $E_w$, we multiplied the result obtained with the Eq. 7 for the number of events, computed

as 365 (days per year) divided by the mean interarrival time at each point within the lagoon. Instead, using the complete

formulation in Eq. 6, the erosion work over one year, $E_w^*$, can be simply computed simply by extending the integration period to the entire year.

We computed the spatial distribution of the annual erosion work $E_w$ using the synthetic approach for the six configurations of the Venice Lagoon (Figure 9). We computed the erosion work also according to Eq. 6, in order to compare differences between the complete formulation based on the computed BSS time records and the synthetic approach exploiting the pos-

395 sibility of describing resuspension events as marked Poisson processes (Figure S15). The erosion work computed following the two approaches is quite similar, as shown by the map of the relative error (Figure S16) and by the computed values of the spatially averaged relative error which varies between 10% and 14% considering all the analyzed configurations of the lagoon (Table 1). Such an agreement between the two estimates of the erosion work supports the validity of the provided statistical characterization of resuspension events.

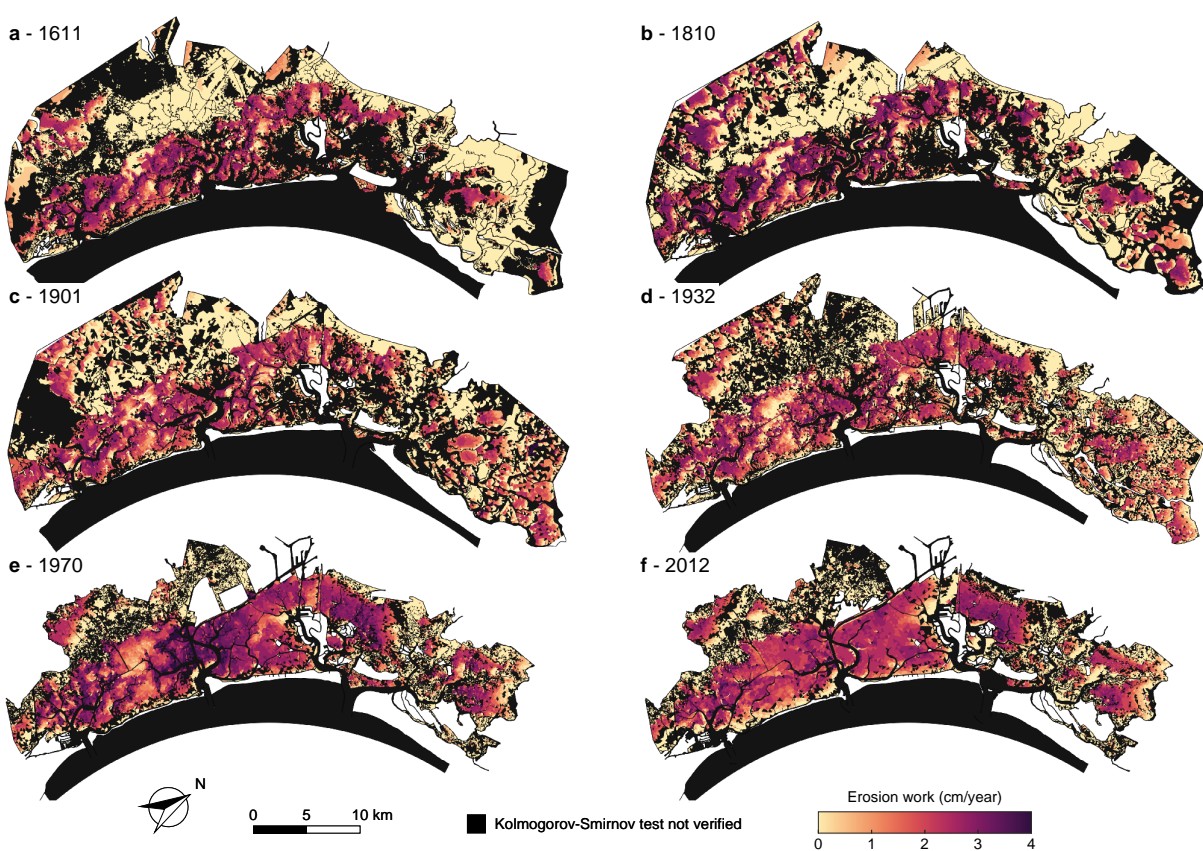

**Figure 9. Erosion work.** Spatial distribution of erosion work for the six different configurations of the Venice Lagoon: (**a**) 1611, (**b**) 1810, (**c**) 1901, (**d**) 1932, (**e**) 1970, and (**f**) 2012. Black identifies sites where the bottom shear stress cannot be modelled as a marked Poisson process (i.e. the KS test is not verified for the interarrival time).

As it is defined, the erosion work represents the total potential erosion, rather than the effective erosion, because it neglects the subsequent possible deposition. However, a comparison between the erosion work and the actual erosion, which can be retrieved from the comparison between surveys, can still provide some interesting insights and highlight erosive trends.

Focusing in particular on the central-southern lagoon, the erosion work is almost constant between 1611 and 1932, it reaches its maximum in 1970 and then it decreases in the present configuration (Figure 9). The four most ancient configurations (i.e. 1611, 1810, 1901 and 1932) display a more complex spatial pattern of the computed erosion work because of the wider presence of salt marshes and islands distributed throughout the basin and because of the shallower and more irregular bathymetry characterizing the tidal flats. This morphology is such that the fetch is continuously interrupted and wind waves are prevented from fully developing while generating and propagating over areas whose bathymetry is continuously varying. Interestingly, even if the present configuration of the lagoon displays larger mean intensities and longer mean durations than in 1970 (see Figure 5 and Figure 6), the combination with generally longer mean interarrival times (Figure 4) affects the erosion work. Indeed, the erosion work is maximum in the 1970 configuration when it reaches a peak of more than 4.0 cm/year. This promoted an intense and uniform erosion of the lagoon, thus leading to the present morphology and bathymetry characterized by less complex erosion patterns and a roughly constant erosion work on the tidal flats in the central-southern lagoon of about 2.5 cm/year. Our results quantitatively support previous studies (Carniello et al., 2009; Molinaroli et al., 2009) that identified two different evolutionary trends in the northern lagoon and in the central-southern part, the northern lagoon displaying, on average, much lower erosion rates.

**Table 1.** Spatially averaged relative error between erosion work computed with Eq. 6 and 7

| Configuration | $r_e$ [-] |
|---|---|
| 1611 | 0.140 |
| 1810 | 0.108 |
| 1901 | 0.135 |
| 1932 | 0.131 |
| 1970 | 0.133 |
| 2012 | 0.112 |

## 4   Conclusions

Our results provide a statistical characterization of sediment erosion dynamics, aimed at testing the possibility to describe erosion events as a Poisson process in a synthetic modelling framework able to reproduce the long-term evolution of shallow tidal systems. The proposed approach aims to better describe erosion events in shallow tidal environments, where BSS dynamics are strongly affected by wind conditions.

In the present study, the approach is applied to the specific case of the Venice Lagoon, for which six morphological configurations along the last four centuries are available. We applied the extensively calibrated and tested Wind Wave-Tidal Model to the

six historical configurations of the Venice Lagoon, in order to perform a spatially-explicit analysis of the BSS time series under the same wind and water level forcing. We analyzed the computed BSS temporal evolution following the peak-over-threshold theory. We verified whether wind-wave erosion events could be modelled as a marked Poisson process by performing the non-parametric Kolmogorov-Smirnov goodness of fit test to confirm the hypothesis that the interarrival time of over-threshold BSS events together with their durations and intensities are exponentially distributed random variables.

Statistical analyses of the wave-driven erosion processes suggest that interarrival times between two consecutive over-threshold events, their durations and intensities can be described as exponentially distributed random variables over wide areas in all the selected configurations of the Venice Lagoon. As a consequence, the wave-induced erosion can be represented by a marked Poisson process through centuries.

Furthermore, we observed that durations and intensities of over-threshold BSS exceedances are highly correlated, while almost no correlation exists between duration and interarrival time, as well as between intensity and interarrival time. These observations indicate that a 3-D Poisson process, in which the marks (duration and intensity of the over-threshold events) are mutually dependent but independent from the interarrival time, provides a suitable description of the wave-induced erosion processes.

Moreover, we showed that in the last four centuries the interarrival times of erosion events generally increased everywhere within the lagoon, as well as their intensities and durations, thus leading to less frequent but more intense wave-induced erosion events. These modifications in the bottom shear stress field are generated by, but at the same time they are also responsible for, the morphological modifications of the Venice Lagoon, in particular the generalized deepening of tidal flats and reduction of salt-marsh area. Only in the "Fondo dei Sette Morti", located close to the watershed divide between the Malamocco and the Chioggia inlets, interarrival times decrease in the last four centuries. Such an opposite trend is associated with the relatively deep and constant bottom elevation characterizing this area combined with the generalized deepening experienced by the surrounding areas that allows more frequent events reaching the "Fondo dei Sette Morti".

The erosion work, computed as a combination of interarrival times, durations and intensities, remained almost constant and characterized by an irregular spatial pattern until the beginning of the twentieth century, when it rapidly increased reaching a peak in 1970. In the last few decades, the erosion work decreased, presenting a more uniform pattern and suggesting that the quite intense erosive trend the Venice Lagoon has been experiencing since the beginning of the last century is, at present, slowing down as a consequence of the generalized deepening and flattening of the lagoonal bed. Owing to the choice of forcing the domain with the same conditions, these changes in the erosive trend are, in fact, only due to morphological modifications experienced by the tidal basin.

The present findings, together with the statistical characterization of suspended sediment dynamics (Tognin et al., 2023), represent a step towards the set up of a synthetic, statistically-based framework which can be used to model the long-term morphodynamic evolution of shallow tidal systems through the use of independent Monte Carlo realizations, thus possibly exploring a large set of equally likely lagoonal configurations.

*Data availability.* All data presented in this study and used for the analysis of the bottom shear stress are available at https://researchdata.cab.unipd.it/id/eprint/728 (10.25430/researchdata.cab.unipd.it.00000728)

*Author contributions.* Conceptualization: Andrea D'Alpaos, Andrea Rinaldo, Luca Carniello, Davide Tognin;

Methodology: Davide Tognin, Andrea D'Alpaos, Luca Carniello;

Formal analysis and investigation: Davide Tognin, Laura Tommasini;

Figures: Davide Tognin;

Writing - original draft preparation: Davide Tognin, Laura Tommasini;

Writing - review and editing: all authors;

Funding acquisition: Andrea D'Alpaos, Luca Carniello;

Resources: Andrea D'Alpaos, Luca Carniello, Luigi D'Alpaos, Andrea Rinaldo;

Supervision: Andrea D'Alpaos, Luca Carniello.

*Competing interests.* The authors declare no competing interests.

*Acknowledgements.* This scientific activity was partially performed within the Research Programme Venezia2021, with contributions from

the Provveditorato for the Public Works of Veneto, Trentino Alto Adige and Friuli Venezia Giulia, provided through the concessionary of State Consorzio Venezia Nuova and coordinated by CORILA, Research Line 3.2 (PI A.D.), the 2019 University of Padova project (BIRD199419) 'Tidal network ontogeny and evolution: a comprehensive approach based on laboratory experiments with ancillary numerical modelling and field measurements' (PI L.C.), the University of Padova SID2021 project, 'Unraveling Carbon Sequestration Potential by Salt-Marsh Ecosystems' (P.I. A. D.), and the iNEST (Interconnected Nord-Est Innovation Ecosystem) project and received funding from the European Union

Next-GenerationEU (National Recovery and Resilience Plan – NRRP, Mission 4, Component 2, – D.D. 1058 23/6/2022, ECS00000043 - CUP: C43C22000340006).

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
