# Peer review of "Statistical characterization of erosion and sediment transport mechanics in shallow tidal environments. Part 1: erosion dynamics"

_EGUsphere, 2023_

## Referee Comment (RC1)

**Comments on "Statistical characterization of erosion and sediment transport mechanics in shallow tidal environments Part 1: erosion dynamics"**

**1 Summary**

This work introduced the idea of using random processes to model the wave-tidal-induced erosion events along the coastal area. The Venice Lagoon, Italy is chosen as the study site due to the availability of multiple bathymetry surveys over the past centuries. The numerical model Wind Wave Tidal Model (WWTM) is used to simulate hydrodynamics conditions, and statistics are extracted from simulation results. The author found that the interarrival time of the erosion events follows an exponential distribution, hence the events can be modeled as marked Poisson process. This work paved a new way to upscale short-term simulations in a wave-tidal environment to long-term, while following the statistical characteristics. This statistical model-based upscale technique is the major scientific contribution of this work. My main comments on this work are on the boundary conditions, that are used in the upscale analysis (erosion work). I believe when dealing with a time scale over 4 centuries, the climate can play an important role, the analysis and boundary conditions should take into consideration of it. There are some details such as knowledge gaps that need to be improved, which are listed below. I also recommend the authors show more details on how the model is set up and validated; show the reasoning behind the choice of several important constants such as critical shear stress and erosion coefficient. Overall, I think this work is of high quality, and it should be rated as a minor revision or on the editor's decision.

**2 Major comments**

1. The knowledge gap is not clear from the literature reviews. The author stated in Line 53 that the main goal of this work is to figure out whether the wave-induced erosion events can be modeled as marked Poisson. However, in previous paragraphs, the author didn't show there is a knowledge gap that requires to verify. For example, is there another study using random processes to model erosion events? What do other researchers usually do, when they need to simulate/upscale long-term morphodynamics in coastal areas? etc.

2. Line 104. It will be nice to show more details in WWTM model, such as showing the equations. For example which form of the shallow equation is solved, how bottom friction and wave energy are simulated, etc. Is there a turbulence closure used in this model?

3. Line 120. Will be more convincing to show the comparison of numerical modeling results in this study and the measured data.

4. Line 126-129. The method is not clear, please rephrase this sentence.

5. Line 130. Thinking about the time scale over 4 centuries, the climate has changed, e.g., the sea level, and mean temperature. Do you think the old morphology is the result of the old climate and the new morphology is the result of the new climate? If so it makes more sense to also consider the climate in the design of the boundary conditions.

6. Line 150. Why choose KS test? There are multiple statistical tests, such as, Anderson-Darling or Cramer Von Mises, etc. Do all these tests give similar results?

7. Line 166. The choice of critical shear stress is very important in this study. Please show more details on how 0.4 Pa is calculated/estimated.

8. Line 166. Is the outcome of this study sensitive to the choice of critical shear stress?

9. Line 270. Is there a way to validate the choice of e in the "erosion work"? Maybe using the differences between these bathymetry data, the "true" erosion rate can be estimated? And then use it to estimate e?

10. Line 292. Again, do you think these complications are due to using a modern climate and ancient bathymetry?

11. Line 324 Similarly, if the sea level is lower in the past, wouldn't that increase the interarrival times? Will this consideration change the increasing trend?

12. Line 289, 305. Since the goal of this study is to upscale short-term simulations, and there are 6 surveys over a long period of time, is it possible to use an older survey and this statistical model to predict a newer survey? If so what does the comparison of the results look like?

**3   Other comments**

1. A typo in the second affiliation "Department of Geosciences ..."

2. In the captions in Figure 2, 3, 5, 6, The description of subfigures is confusing. Recommend switching the order of the year and sub-figure numbering. For example, use "(a) 1611; (b) 1810; ..." instead.

---

## Author Comment (AC1)

**Author Response to Reviews of Earth Surface Dynamics Manuscript egusphere-2023-319**

**Statistical characterization of erosion and sediment transport mechanics in shallow tidal environments. Part 1: erosion dynamics**

Andrea D'Alpaos[1], Davide Tognin[1,2], Laura Tommasini[1], Luigi D'Alpaos[2], Andrea Rinaldo[2,3], and Luca Carniello[2]

[1] *Department of Geosciences, University of Padova, Padova, Italy*
[2] *Department of Civil, Environmental, and Architectural Engineering, University of Padova, Padova, Italy*
[3] *Laboratory of Ecohydrology ECHO/IEE/ENAC, Ècole Polytechnique Fèdèrale de Lausanne, Lausanne, Switzerland*
Correspondence: Davide Tognin (davide.tognin@unipd.it)

**Summary**

The authors are grateful to the editorial board and the reviewers for their thoughtful and constructive comments on our paper, which significantly improved the manuscript and how our findings are communicated.
Following the Reviewers' suggestions, we have carefully revised the introduction in order to better highlight how the proposed approach aims to contribute to filling the knowledge gap in long-term morphodynamic modelling and to better frame the potential applicability of this approach.
Moreover, the revised manuscript now includes a more detailed description of the numerical hydrodynamic model used in our analysis and its calibration procedure.
Finally, as suggested by the Reviewers, we provided additional details about some modelling choices that were not properly justified in the previous version of the manuscript. In particular, now we extensively discuss the choice of boundary conditions and the threshold shear stress to apply the peak-over-threshold analysis.
Overall, in the new version of the manuscript, we consistently revised the main text and importantly expanded the Supplementary Information, by adding the detailed model description and figures S2 to S6.
In the following, we discuss in detail all Reviewers' comments and show how we addressed them, referencing line numbers in the revised version of the manuscript with the track changes.
Please note that the Reviewers' comments are in blue, our detailed responses are in black, and the text of the manuscript is framed.

*Legend*

RC:     Reviewer Comment

AR:     Author Response

☐ :     Modified manuscript text

*Note*: References to reviewers' comments are indicated as RCx.x and numbered progressively.

**Reply to Reviewer #1**

RC1.0: This work introduced the idea of using random processes to model the wave-tidal-induced erosion events along the coastal area. The Venice Lagoon, Italy is chosen as the study site due to the availability of multiple bathymetry surveys over the past centuries. The numerical model Wind Wave Tidal Model (WWTM) is used to simulate hydrodynamics conditions, and statistics are extracted from simulation results. The author found that the interarrival time of the erosion events follows an exponential distribution, hence the events can be modeled as marked Poisson process. This work paved a new way to upscale short-term simulations in a wave-tidal environment to long-term, while following the statistical characteristics. This statistical model-based upscale technique is the major scientific contribution of this work. My main comments on this work are on the boundary conditions, that are used in the upscale analysis (erosion work). I believe when dealing with a time scale over 4 centuries, the climate can play an important role, the analysis and boundary conditions should take into consideration of it. There are some details such as knowledge gaps that need to be improved, which are listed below. I also recommend the authors show more details on how the model is set up and validated; show the reasoning behind the choice of several important constants such as critical shear stress and erosion coefficient. Overall, I think this work is of high quality, and it should be rated as a minor revision or on the editor's decision.

AR: We thank the Reviewer for his/her positive comments on our manuscript and for his/her insightful suggestions that contributed to improving the quality and clarity of our manuscript. Please, find in the following the responses to each detailed comment.

RC1.1: The knowledge gap is not clear from the literature reviews. The author stated in Line 53 that the main goal of this work is to figure out whether the wave-induced erosion events can be modeled as marked Poisson. However, in previous paragraphs, the author didn't show there is a knowledge gap that requires to verify. For example, is there another study using random processes to model erosion events? What do other researchers usually do, when they need to simulate/upscale long-term morphodynamics in coastal areas? etc.

AR: We agree with the Reviewer that the knowledge gap was not particularly clear. Following Reviewer's suggestion, we deeply revised the introduction as follows:

> (line 50) ~~The wind-wave and the related BSS fields, as well as their impact on the morphodynamics of shallow tidal basins, can be provided by several numerical models. However, modelling the morphodynamic evolution over time scales of centuries using fully-fledged models is a difficult task due to the computational burden involved and, therefore, simplified approaches are more and more frequently adopted (Murray, 2007). Opting for such a simplified approach, long-term models ideally should be tested in tidal systems 
[revised manuscript text omitted]

Moreover, to better stress the aim and the logical structure of the approach, we also revised the abstract as follows:

(line 1)  Reliable descriptions of erosion events are foundational to effective frameworks relevant to the fate of tidal landscape evolution. *Besides the rhythmic, predictable action of tidal currents, erosion in shallow tidal environments is strongly influenced by the stochastic wave-induced bottom shear stress (BSS), mainly responsible for sediment resuspension on tidal flats.* However, the absence of *sufficiently long* measured time series of *BSS* prevents a direct analysis of *the combined tide- and wave-driven* erosion dynamics *and its proper representation in long-term morphodynamic models. Here we test the hypothesis of describing erosion dynamics in shallow tidal environments as a Poisson process by analysing with the peak-over-threshold theory the BSS time series computed using a fully-coupled, bi-dimensional numerical model. We perform this analysis on the Venice Lagoon, Italy, taking advantage of the availability of several historical surveys in the last four centuries, which allow us to investigate the effects of morphological modifications on spatial and temporal erosion patterns. Our analysis suggests that erosion events on intertidal flats can*

*effectively be modelled as a marked Poisson process in different morphological configurations, because interarrival times, durations and intensities of the over-threshold exceedances are always well described by exponentially distributed random variables.* ~~Here we adopted a fully-coupled, bi-dimensional numerical model to compute BSS generated by both tidal currents and wind waves in six historical configurations of the Venice Lagoon in the last four centuries. The one-year-long time series of the total BSS were analyzed based on the peak-over-threshold theory to statistically characterize events that exceed a given erosion threshold and investigate the effects of morphological modifications on spatial and temporal erosion patterns. Our analysis suggests that erosion events can be modelled as a marked Poisson process in the intertidal flats for all the considered configurations of the Venice Lagoon, because interarrival times, durations and intensities of the over-threshold exceedances are well described by exponentially distributed random variables. Moreover, while the intensity and duration of over-threshold events are temporally correlated, almost no correlation exists between them and interarrival times.~~ The resulting statistical characterization allows a straightforward computation of morphological indicators, such as the erosion work, and paves the way to a novel synthetic, yet reliable, approach for the long-term morphodynamic modelling of tidal environments.

RC1.2: Line 104. It will be nice to show more details in WWTM model, such as showing the equations. For example which form of the shallow equation is solved, how bottom friction and wave energy are simulated, etc. Is there a turbulence closure used in this model?

AR:  When presenting results computed with an already-published model, finding a satisfactory compromise between the will of making each contribution a stand-alone work and the need to maintain the manuscript concise without repeating already published details is not an easy task.

As highlighted by the Reviewer's summary comment, the major scientific contribution of this work is the upscaling technique based on the statistical characterization of erosion events. Therefore, we believed that WWTM could not be considered the main focus of the paper, even though it is functional for the BSS computation, which is a necessary step for statistical analysis because BSS time series are hardly available from direct field measurements.

Thanks to the Reviewers' comments, we realized that the summary we provided in the first version of the manuscript was lacking some necessary details, such as a precise description of how the model computes the BSS, which is indeed crucial for the subsequent analysis. For this reason, we included in the method section the equations implemented in the WWTM model to compute the different contributions to BSS, by modifying the text as follows:

(line 164) The bottom shear stress induced by currents, $\tau_{tc}$, is evaluated using the Strickler equation considering the case of a turbulent flow over a rough wall, *which can be written as (Defina, 2000)*

$$\tau_{tc} = \rho\, g\, Y \left( \frac{|\boldsymbol{q}|}{K_s^2 H^{10/3}} \right) \boldsymbol{q}$$

*where $\rho$ is water density, $g$ is the gravity acceleration, $Y$ is the effective water depth, defined as the volume of water per unit area actually ponding the bottom, $\boldsymbol{q}$ is the flow rate per unit width, $K_s$ is the Strickler roughness coefficient, and $H$ is an equivalent water depth that accounts for the typical height of ground irregularities.*

(line 176) The wind-wave module computes the bottom shear stress induced by wind waves as (*Carniello, 2005*)

$$\tau_{ww} = \frac{1}{2}\rho\, f_w\, u_m^2$$

*where $u_m$ is the maximum horizontal orbital velocity associated with wave propagation and $f_w$ is the wave friction factor. According to the linear theory, the bottom velocity $u_m$ can be evaluated as*

$$u_m = \frac{\pi H_w}{T\, \sinh(kh)}$$

*where $H_w$ is the wave height, $T$ denotes the wave period, $k$ is the wave number, and $h$ is the water depth. The wave friction factor can be approximated as (Soulsby, 1997)*

$$f_w = 1.39\left[\frac{u_m T}{2\,\pi(D_{50}/12)}\right]^{-0.52}$$

*where $D_{50}$ is the median grain size.*

(line 186) The total bottom shear stress, $\tau_{wc}$, resulting from the combined effect of tidal currents and wind waves, is enhanced beyond the sum of the two contributions, because of the non-linear interaction between the wave and the current boundary layer. In the WWTM this is accounted for by using the empirical formulation suggested by Soulsby (1995, 1997):

$$\tau_{wc} = \tau_{tc} + \tau_{wc}\left[1 + 1.2\left(\frac{\tau_{ww}}{\tau_{ww}+\tau_{tc}}\right)\right]$$

Instead, reporting in the manuscript extensive, already-published details regarding the model's basic equations would be redundant and unnecessarily lengthen the manuscript compared to the little benefit for the reader. Therefore, we deem that these details, such as the form of the shallow water equations solved, the wave energy equation and the turbulence closure model, would better fit the supplementary material. We report here for the Reviewer's convenience the text we added as Supplementary Information:

***Supplementary Information***

***Hydrodynamic model***
*The hydrodynamic module solves the 2D depth-integrated shallow water equations, phase averaged over a representative elementary area in order to deal with wetting and drying processes in very shallow and irregular domains (Defina, 2000):*

$$\vartheta(\eta)\frac{\partial\eta}{\partial t} + \nabla\cdot\boldsymbol{q} = 0$$
$$\frac{D}{Dt}\left(\frac{\boldsymbol{q}}{Y}\right) + \frac{1}{Y}\nabla\cdot\boldsymbol{Re} + \frac{\tau_t}{\rho Y} - \frac{\tau_s}{\rho Y} + g\,\nabla h = 0$$

*where $t$ is time, $D/Dt$ is the material time derivative, $\nabla$ and $\nabla\cdot$ denote the 2D gradient and divergence operators, respectively. In the continuity equation (Eq. 1), $\eta$ is the free surface elevation, $\vartheta$ is the wet fraction of the computational domain,*

*which is a function of water depth and local topographic unevenness (Defina, 2000), $\mathbf{q} = (q_x, q_y)$ is the depth-integrated velocity (i.e., discharge per unit width). In the momentum equations (Eq. 2), Y is the effective water depth (i.e., the water volume per unit area), $\boldsymbol{\tau_t}$ and $\boldsymbol{\tau_s}$ are the shear stresses at the bottom and at the free surface, respectively, $\rho$ is water and g is gravity acceleration.*

*The Reynolds stresses $\boldsymbol{Re}$ are computed using a depth-averaged version of Smagorinsky's model (Smagorinsky, 1963) and they read:*

$$\boldsymbol{Re} = R_{ij} = \nu_e \, Y \left( u_{i,j} + u_{j,i} \right)$$

*with $i, j$ denoting either the x or y coordinate, $\boldsymbol{u} = \boldsymbol{q}/Y$ and the eddy viscosity $\nu_e$ is computed as*

$$\nu_e = 2 \, C_S^2 \, A_e \, \sqrt{2\left(u_{x,x}\right)^2 + \left(u_{x,y} + u_{y,x}\right)^2 + 2\left(u_{y,y}\right)^2}$$

*where the Smagorinsky coefficient $C_S$ is set equal to 0.2 and $A_e$ is the area of the computational element.*

*In the numerical scheme, the material time derivative in Eq.2 is solved with the method of characteristics by expressing it as the finite difference in time. This allows solving the continuity equation (Eq. 1) with a semi-implicit scheme, resulting in a self-adjoint spatial operator, which is solved on a staggered triangular grid with the finite element method of Galerkin and flow rates are obtained by back substitution.*

**Wind wave model**

*The wind-wave module (Carniello et al., 2011) solves the wave action conservation equation using the same computational grid of the hydrodynamic module, which provides water depths and depth-averaged flow velocities, used to propagate the wind-wave field. In the frequency domain, the wave action density, $N_0$, evolves according to (Carniello et al., 2005)*

$$\frac{\partial N_0}{\partial t} + \frac{\partial}{\partial x} c'_{gx} \, N_0 + \frac{\partial}{\partial y} c'_{gy} \, N_0 = S_0$$

*where $c'_{gx}$ and $c'_{gy}$ are the wave group celerity components used to approximate the speed propagation of $N_0$ (Holthuijsen et al., 1989}, $S_0$ denotes the wind-wave source terms, accounting both for positive (wind energy input) and negative (bottom friction, whitecapping, and depth-induced breaking) contributions. The wave-action conservation equation (Eq. 3) is solved with an upwind finite volume scheme based on the same computational grid of the hydrodynamic model. In each element at each time step, the wind-wave model computes the wave action, from which the significant wave height is obtained by applying the linear theory.*

We deem that this compromise provides the necessary information to understand the adopted computational tool without lengthening too much the manuscript.

RC1.3: Line 120. Will be more convincing to show the comparison of numerical modeling results in this study and the measured data.

AR:  As for the equations, we believe that also for the comparison of numerical results with measured data, we need to keep in mind the readability of the manuscript. Reporting a detailed description of the model performance against measured data (already published in previous contributions) in the main text would importantly lengthen the manuscript and undermine its readability, therefore, we deem this is not the best solution to be adopted.

Anyhow, to achieve a reasonable compromise between readability and the Reviewer's suggestion, we added a paragraph on the model performance in the Method section, to provide the reader with more information on the capability of the model to capture the hydrodynamic processes at hand, leaving additional Figures showing a detailed comparison between numerical modelling and measured data in the Supplementary Information.

The revised version of the text now reads:

> (line 192) The WWTM has been widely tested against field observations not only in the Venice Lagoon (e.g., Carniello et al., 2005, 2011; Tognin et al., 2022) but also in other shallow microtidal environments worldwide, for example in the back-barrier lagoons of the Virginia Coast Reserve (Mariotti et al., 2010) and the Cádiz Bay (Zarzuelo et al., 2018, 2019). *Concerning the Venice Lagoon, model calibration and testing have been performed only in the most recent configuration, i.e., when field data are available. For the older configurations of the lagoon, no hydrodynamic measurements are available and, consequently, the roughness coefficient values have been derived in analogy from those selected for the calibrated grid in the most recent configuration, by comparing local sediment grain size, bed elevation and morphological classes (e.g., channel, tidal flat, salt marsh), which also take into account for the possible presence of vegetation (Finotello et al., 2023).*
>
> *We summarise here the model performance in reproducing tidal levels, significant wave heights and flow rates at the inlets by reporting the standard Nash-Sutcliffe Model Efficiency (NSE) parameter computed when field data are available and refer the interested reader to the Supplementary Information (Figures S2 to S5) and the literature (Carniello et al., 2005, 2011; Tognin et al., 2022) for a more detailed comparison. Adopting the classification proposed by Allen et al. (2007), the model performance can be rated from excellent to poor (i.e., NSE > 0.65 excellent; 0.5 < NSE < 0.65 very good; 0.2 < NSE < 0.5 good; NSE < 0.2 poor). The WWTM model is excellent in reproducing tidal levels ($NSE_{mean}$ = 0.970, $NSE_{median}$ = 0.984, $NSE_{std}$ = 0.040), very good to excellent in reproducing significant wave heights ($NSE_{mean}$ = 0.627, $NSE_{median}$ = 0.756, $NSE_{std}$ = 0.357), and excellent in replicating flow rates at the inlets ($NSE_{mean}$ = 0.853, $NSE_{median}$ = 0.931, $NSE_{std}$ = 0.184) (Statistics are derived from calibration reported in Carniello et al. (2011), their Tables 1,2, and 3, and Tognin et al. (2022), their Table S2).*

For the sake of brevity, we do not report here the text and figures added in the Supplementary information to show the result of the calibration procedure.

RC1.4: Line 126-129. The method is not clear, please rephrase this sentence.

AR:    We rephrased this sentence as follows:

> (line 209) We applied the numerical model to the six computational domains representing the Venice Lagoon and a portion of the Adriatic Sea in front of it in order to perform one-year-long simulations (Figure S1). The boundary conditions of the model are the hourly tidal levels measured at the Consiglio Nazionale delle Ricerche (CNR) Oceanographic Platform, located in the Adriatic Sea *approximately 15 km offshore of the coastline*, and wind velocities and directions recorded at the Chioggia anemometric station, for which a quite long data set was available (Figure 1a).
>
> *In particular, we selected as boundary conditions measurements recorded in 2005, because they can be considered representative of the wind climate of the Venice Lagoon. Indeed, the probability distribution of wind speeds in 2005 shows the minimum difference with the mean probability distribution computed for the period 2000-2020, compared to any other year within the same time interval (Supplementary information, Figure S6 and Table S1). Indeed, a visual comparison between the wind roses for 2005 and the entire period 2000-2020 supports this choice (Figure 1b).*

To better explain this choice, we also added further information in the Supplementary information as follows:

> **Supplementary Information**
>
> *To simulate the typical wave-driven BSS conditions in the Venice Lagoon, we carefully analyse the wind climate measured in the period 2000-2020. In particular, we compared the wind velocity measured in each year with that of the whole period 2000-2020 by means of the two sample Kolmogorov-Smirnov (KS) test and the Wilcoxon (W) test (Table S1). We selected the wind velocity measured in 2005 because it does not significantly differ from that of the whole considered period according to both the KS and W tests. A visual comparison of the cumulative distribution frequency F(x) of wind velocity also confirms this choice (Figure S6).*

[Figure]

**Figure S6. Wind climate for the period 2000-2019.** Cumulative density function F(x) of wind velocity for the period 2000-2020. Grey lines represent F(x) of every single year, the red dashed line is the year 2005, and the black thick line represents the whole period 2000-2020.

**Table S1.** Results of the Kolmogorv-Smirnov (KS) and Wilcoxon (W) tests on wind velocity.

| Year | KS test | | W test | |
|------|---------|---|--------|---|
| | p-value | h | p-value | h |
| 2000 | < 0.001 | 1 | < 0.001 | 1 |
| 2001 | < 0.001 | 1 | < 0.001 | 1 |
| 2002 | < 0.001 | 1 | < 0.001 | 1 |
| 2003 | < 0.001 | 1 | < 0.001 | 1 |
| 2004 | < 0.001 | 1 | 0.024 | 1 |
| **2005** | **0.072** | **0** | **0.283** | **0** |
| 2006 | < 0.001 | 1 | < 0.001 | 1 |
| 2007 | 0.002 | 1 | 0.712 | 0 |
| 2008 | < 0.001 | 1 | < 0.001 | 1 |
| 2009 | 0.002 | 1 | 0.006 | 1 |
| 2010 | < 0.001 | 1 | < 0.001 | 1 |
| 2011 | < 0.001 | 1 | < 0.001 | 1 |
| 2012 | < 0.001 | 1 | < 0.001 | 1 |
| 2013 | < 0.001 | 1 | < 0.001 | 1 |
| 2014 | < 0.001 | 1 | 0.003 | 1 |
| 2015 | < 0.001 | 1 | 0.020 | 1 |
| 2016 | < 0.001 | 1 | 0.055 | 0 |
| 2017 | < 0.001 | 1 | 0.033 | 1 |
| 2018 | < 0.001 | 1 | < 0.001 | 1 |
| 2019 | < 0.001 | 1 | < 0.001 | 1 |
| 2020 | < 0.001 | 1 | < 0.001 | 1 |

 Thinking about the time scale over 4 centuries, the climate has changed, e.g., the sea level, and mean temperature. Do you think the old morphology is the result of the old climate and the new morphology is the result of the new climate? If so it makes more sense to also consider the climate in the design of the boundary conditions.

AR: We thank the Reviewer for the comment, which helped us to better explain some modelling choices and, thus, to improve the clarity of our manuscript.

First of all, relative sea level changes surely play a fundamental role in shaping the lagoon morphology. Indeed, we implicitly accounted for sea level changes, because each bathymetrical survey and, hence, the correspondent elevation of the computational grid was referred to the coeval mean sea level. Thanks to the Reviewer's comment we realized that this important concept was not properly described in the original version of the manuscript and, therefore, we better highlighted it in the revised version, as follows:

> (line 151) *In each bathymetry and, hence, in each computational grid, bed elevation refers to the local mean sea level at the time when each survey was performed.*
>
> (line 221) *Because bed elevation in each computational grid refers to the coeval mean sea level, by using the same water levels as boundary conditions we implicitly take into account the effect of historical relative sea level variations.*

To answer the more general question about the relationship between climate and morphology, we clarify that, although climate changed in the last four centuries, human interventions undeniably played a primary role in affecting the morphological changes in the same period (Carniello et al., 2009; D'Alpaos, 2010a, 2010b; Finotello et al., 2023; Silvestri et al., 2018). A compelling example is provided by a visual comparison of the bathymetries represented in Figure 2. For instance, the morphological changes that occurred between 1932 and 1970 (i.e., less than 40 years) cannot be only the result of climate change, however sudden this change could have been. Therefore, we can conclude that the effects of changes in climate on the morphology of the Venice Lagoon are small compared to those resulting from human interventions.

In the revised version of the manuscript, we highlighted that human interventions played a prominent role in the morphological evolution compared to the change in climates by modifying the text as follows:

> (line 127) *The morphology of the Lagoon deeply changed over the last four centuries (Figure 2), especially owing to anthropogenic modifications (Carniello et al. 2009; D'Alpaos, 2010, Silvestri et al. 2018; Finotello et al. 2023).*

For the sake of brevity, we do not report here the description of the interventions, that was already provided in the first version of the manuscript.

> (line 137) *As a result, these human interventions, together with eustatic sea-level rise (average value 1.23 ± 0.13 mm/year between 1872 and 2019; 2.76± 1.75 mm/year between 1993 and 2019; see Zanchettin et al., 2021), played a primary role in affecting the morphological evolution of the lagoon.*

Besides these useful clarifications, we must also stress that, rather than trying to reconstruct the exact climate forcing that gave rise to the present-day morphology, the aim of this study is to understand how morphological changes affect the parameters driving erosive processes. From this point of view, the different historical configurations can be considered as different possible morphologies of the same area (to which we refer using the year only for the sake of simplicity). But, more importantly, setting the same boundary conditions is necessary to

highlight the specific role of the morphology in driving the hydrodynamic and wave fields and, therefore, the erosive process. This is indeed the power of numerical modelling which enables one to investigate scenarios otherwise impossible to get in reality. If we would have used the tidal and meteorological conditions to which the ancient configurations of the lagoon were actually subjected (although clearly impossible due to data unavailability), it would have been impossible to distinguish the effect of the morphology and those associated to the different boundary conditions based on model results (BSS).

We have better highlighted this aspect in several points of the revised manuscript:

> (line 10) *We perform this analysis on the Venice Lagoon, Italy, taking advantage of the availability of several historical surveys in the last four centuries, which allow us to investigate the effects of morphological modifications on spatial and temporal erosion patterns. Our analysis suggests that erosion events on intertidal flats can effectively be modelled as a marked Poisson process in different morphological configurations, because interarrival times, durations and intensities of the over-threshold exceedances are well described by exponentially distributed random variables.*
>
> (line 222) *Considering the same wind and tidal forcing in each historical configuration of the Venice Lagoon allowed us to isolate the effects of the morphology on the hydrodynamics and wind-wave fields*
>
> (line 381) *A punctual comparison among different configurations provide further insight into the effects of morphological changes on interarrival times (Figure 8). On a tidal flat in the northern lagoon named ``Palude Maggiore'' (see station S1 in Figure 1a), as in most areas of the lagoon, the mean interarrival time $\bar{\lambda}_t$ between two subsequent over-threshold events increases through time (Figure 8a). This is because this area preserved the same morphological features, i.e. relatively shallow tidal flats protected by the mainland and salt marshes, over the last four centuries.*

RC1.6: Line 150. Why choose KS test? There are multiple statistical tests, such as, Anderson-Darling or Cramer Von Mises, etc. Do all these tests give similar results?

AR: Following the Reviewer's suggestion we further investigated this issue. Among the plethora of parametric and non-parametric goodness-of-fit tests, we focused on the empirical distribution function (EDF) statistics – such as Kolmogorov-Smirnov (KS), Kuiper (V), Cramer von Mises ($W^2$) Anderson-Darling ($A^2$) test - because they generally give more powerful tests than the classical $X^2$ (Stephens, 1974). Various studies in the literature analyse the power of EDF statistics in the case of a normal distribution (D'Agostino, 1986; Stephens, 1974; Thode, 2002). When testing normality, the $A^2$ test was found to perform slightly better than the $W^2$ and KS tests (D'Agostino, 1986). However, this result cannot be regarded as general because it is highly dependent on the symmetry and type of the distribution (Thode, 2002). Indeed, in our case, we test an exponential distribution, not a normal one.

Moreover, as reported in many statistics textbooks, the choice of a goodness-of-fit test cannot be based solely on the power of a statistics, but also on the ease or practicality of computation and availability of necessary tables (coefficients and critical values). From this point of view, the KS test has the advantage of not depending on the specific distribution being tested. On the contrary, the $A^2$ test makes use of specific distribution in calculating critical values for each distribution being tested and, therefore, specific tables are required.

Following the Reviewer's suggestion, to verify our results, we performed also the $A^2$ test on interarrival times, intensity and duration of over-threshold erosion events. We limit our comparison to KS and $A^2$ tests to use built-in functions in Matlab (Matwoks, 2023) and facilitate the reproducibility of our results, avoiding custom-built functions.

In general, the results of the $A^2$ test show little or no difference from those of the KS test, independently of the morphological configuration of the Venice Lagoon. However, in many computational elements, even though the $A^2$ test was computed, it could not be performed properly because of the lack of tabulated values for the exponential distribution in the built-in function (the critical value was automatically substituted with the nearest tabulated value available). Limiting the comparison to the properly calculated values, the $A^2$ and KS tests still show little to no difference.

To summarize, we decided to present the results of the KS tests because of the following reasons:

- considering the power of the test, actually, there is no specific reason to choose one test rather than another when testing exponential distributions;
- in general, the applicability of the $A^2$ test is limited by the availability of specific tables, but, in our specific case, when both $A^2$ and KS can be properly performed they show little to no difference;
- the similar analysis presented by D'Alpaos et al. (2013) for the present-day configuration of the Venice Lagoon makes use of the KS test and we believe that using the same test would help the reader to compare the results.

RC1.7: Line 166. The choice of critical shear stress is very important in this study. Please show more details on how 0.4 Pa is calculated/estimated.

RC1.8: Line 166. Is the outcome of this study sensitive to the choice of critical shear stress?

AR:     We decided to merge together our responses to RC1.7 and 1.8 because these two observations are intimately related to each other.

We understand the Reviewer's point of view and we recognize that, in the first version of the manuscript, the choice of the critical shear stress and its implications were too condensed and, therefore, would have benefited from a more detailed comment. As noted by RC2.8, this additional explanation better suits the method section, so we moved this paragraph from the Result section to the Method section, where the revised version now reads:

> (line 232) In this work, at any location within each considered configuration of the Venice Lagoon, we used the peak-over-threshold (POT) theory (Balkema and de Haan, 1974) to analyze the temporal and spatial evolution of the total BSS, $\tau_{wc}$.
>
> *In general, the selection of the threshold for the POT method must satisfy two contrasting requirements. On the one hand, the threshold must be large enough to discern stochastic events from the deterministic background. On the other hand, the threshold should not be too high to avoid the loss of important information and the need for a much longer time series to compute meaningful statistics, because of the lower number of threshold exceedances. Moreover, the extreme value theory postulates the general emergence of Poisson processes whenever the censoring threshold is high enough (Cramér and Leadbetter, 1967). To comply with these requirements, in the present analysis, the threshold is maintained well below the maximum observed values, in order to remove only the background modulation induced by tidal currents without losing significant information on the stochastic wave-driven erosion process.*

*In applying the POT method to BSS time series, setting the threshold equal to a critical BSS for erosion, $\tau_c$, presents the non-trivial advantage of preserving also the physical meaning of the erosion mechanism. Values of critical BSS for erosion for fine, cohesive mixtures typical of shallow tidal settings largely vary in the literature and are affected by multiple physical and biotic factors (Mehta et al., 1989). Erosion shear stress from in-situ measurements on the tidal flats of the Venice Lagoon ranges between 0.2 and 2.3 Pa (0.7 ± 0.5 Pa - median ± standard deviation), with values higher than 0.9 Pa usually recorded within densely vegetated patches (Amos et al., 2004). In the present analysis, we cannot take into account the role of the biotic component, because of the impossibility to reconstruct the spatial distribution of vegetated tidal flats in the ancient configurations of the Venice Lagoon. For all the above reasons and following the approach suggested by D'Alpaos et al. (2013), we set the critical shear stress, $\tau_c$, equal to 0.4 Pa for all the selected historical configurations of the Venice Lagoon.*

*Before performing the POT analysis, the time series of BSSs were processed by applying a moving average filter, in order to remove spurious upcrossings and downcrossings of the prescribed threshold. This low-pass filter with a time window of 6 hours removes short-term fluctuations, preserving the modulation given by the semidiurnal tidal oscillation. Thanks to this preprocessing procedure, over-threshold events satisfy the independence assumption required by the statistical analysis applied.*

Concerning the sensitivity analysis, we added a paragraph as follows:

*(line 281) The result of modelling erosion events as a Poisson process stands regardless of the specific value of the censoring threshold selected for the POT analysis, provided that it is high enough to exclude deterministic exceedances, and this is confirmed also by the sensitivity analysis performed by D'Alpaos et al. (2013) on the present-day configuration of the Venice Lagoon. Indeed, when considering too low values of the threshold (e.g., $\tau_c$ = 0.2 Pa), deterministic exceedances driven by tidal currents occur and make the interarrival time not exponentially distributed. On the contrary, as the threshold value increases (e.g., $\tau_c \geq 0.6$ Pa), the KS test is still verified, thus confirming that the process remains Poisson for increasing censoring thresholds (see Figure 6 in D'Alpaos et al. (2013) for further details).*

For the Reviewer's convenience, we report here Figure 6 from D'Alpaos et al. (2013) showing the results of the KS test using different BSS thresholds.

[Figure]

Figure 1. Spatial distribution of Kolmogorov-Smirnov (KS) test at a significance level (α = 0.05) on interarrival times assuming a threshold value for the shear stress equal to: $\tau_c = 0.2$ Pa (upper panel), $\tau_c = 0.4$ Pa (central panel) and $\tau_c = 0.6$ Pa (lower panel) (from D'Alpaos et al., 2013)

RC1.9: Line 270. Is there a way to validate the choice of e in the "erosion work"? Maybe using the differences between these bathymetry data, the "true" erosion rate can be estimated? And then use it to estimate e??

AR:     We appreciate the Reviewer's insightful observation on this issue because this is exactly one of the first validation steps that, in principle, can be done once the synthetic model resulting from the statistical characterization of erosion and resuspension dynamics will be applied. However, there is a detail, which may have not been captured because of the lack of clarity in the first version of the manuscript, that makes this validation not possible at the present stage. This detail was hidden in the definition of erosion work. To avoid possible misunderstandings, we better clarified the definition of erosion work in the text as follows:

> (line 412) *In order to provide a more quantitative estimation of the spatial heterogeneity of interarrival times, duration and intensities of the critical BSS exceedances, we computed the "erosion work", which represents the total amount of sediment*  *resuspended during a selected time interval and, thus, the potential erosion, because it does not consider any possible subsequent deposition. The erosion work $[E_w^*]$ experienced by a single point during the time interval $(t_2 - t_1)$ can be computed as:*

$$\left| [E_w^*] \right| = \int_{t_1}^{t_2} \frac{r}{\rho_b} \left( \frac{\tau_{wc} - \tau_c}{\tau_c} \right) dt$$

*where e is the value of the erosion coefficient which depends on the sediment properties and $\rho_b = \rho_s (1 - n)$ is the sediment bulk density, being $n$ the porosity.*  *We set $\rho_s = 2650$ kg m⁻³ and n = 0.4 in agreement with Carniello et al. (2012). In principle, when computing the actual erosion, thus, taking into account both erosive and depositional processes, the parameter "e" could be calibrated by comparing the modelled erosion with that retrieved from the comparison among subsequent surveys provided that other non-natural processes (e.g., boat wave, dredging) do not strongly affect the local morphological evolution. However, because here we are considering the total potential rather than the actual erosion, such a calibration would not be correct. For this reason, we set e equal to $5 \cdot 10^{-5}$ kg m⁻² s⁻¹, as usually suggested for sand-mud mixtures (van Ledden et al., 2004; Le Hir et al., 2007).*

*(line 439) As it is defined, the erosion work represents the total potential erosion, rather than the effective erosion, because it neglects the subsequent possible deposition. However, a comparison between the erosion work and the actual erosion, which can be retrieved from the comparison between surveys, can still provide some interesting insights and highlight erosive trends.*

As it should be clear from the revised text, there is a subtle difference between the erosion work, as we define it, and the "true" erosion as it can be estimated by comparing two surveys, which would have made it not correct to directly perform this comparison. Being defined as the integral of all the over-threshold events, the erosion work represents the total potential erosion and thus completely disregards the possible settling of sediment carried in suspension, once the hydrodynamic conditions are favourable to deposition. Instead, the "true" erosion that can be retrieved by comparing two surveys necessarily takes into account the subsequent deposition, thus differing from the potential erosion computed using the erosion work.

This is the reason why we preferred to talk about "erosive trend" instead of quantitatively comparing the potential and the actual erosion by performing a (wrong) calibration of the parameter "e".

To conclude, to correctly perform the analysis suggested by the Reviewer, two further steps would still be required: 1) the estimation of the potential deposition of suspended sediment, which is the topic of the companion paper; 2) the set up of the stochastic model that considers both erosion and suspended sediment dynamics. It should be clear now that this detailed analysis is beyond the aim of this work and all these elements cannot fit into one single journal article.

**RC1.10: Line 292. Again, do you think these complications are due to using a modern climate and ancient bathymetry?**

AR: We understand the Reviewer's concern. However, as already reported in the response to RC1.5, when considering that this study aims to understand how morphology affects the erosive processes, rather than to reconstruct the hydrodynamics of the historical configurations of the Venice Lagoon, these outcomes are the effects of different morphologies on the hydrodynamics and wind-wave field, instead of resulting from the application of a modern climate to an ancient bathymetry.

RC1.11: Line 324 Similarly, if the sea level is lower in the past, wouldn't that increase the interarrival times? Will this consideration change the increasing trend?

AR:     As reported in the response to RC1.5, in the revised text we better clarified that the different morphological configurations refer to the mean sea level at the time of each specific survey and, hence, we implicitly account for the effects of sea level rise in the last four centuries. Thus, this increasing trend is already the result of considering a lower sea level in the past.

RC1.12: Line 289, 305. Since the goal of this study is to upscale short-term simulations, and there are 6 surveys over a long period of time, is it possible to use an older survey and this statistical model to predict a newer survey? If so what does the comparison of the results look like?

AR:     This is indeed a very interesting comment. Once the theoretical framework is verified (which is indeed the aim of this and of the companion paper) and the statistical model will be set up, one of the most interesting applications, for sure, will be to apply the statistical model to an older surveyed bathymetry of the lagoon and compare model results in term of bed evolution with a more recent surveyed bathymetry, as insightfully suggested by the Reviewer.

However, as we have now better highlighted in the text (see our response to RC1.5), the morphological evolution of the Venice Lagoon has been deeply affected by human interventions over the last centuries and, thus, very likely the comparison between the result of the model and one of the following lagoon configurations will highlight the effects of the anthropogenic interventions (i.e., excavation of large navigable channels, etc), which are not described by the statistical model.

We deem that this topic cannot be discussed within this paper because too many elements for the set-up of the statistical model still need to be presented but, in this comment, the Reviewer precisely captured the global idea motivating our study, of which this contribution is a fundamental step.

RC1.13: A typo in the second affiliation "Department of Geosciences ..."

AR:     Corrected. We thank the Reviewer for noting it.

RC1.14: In the captions in Figure 2, 3, 5, 6, The description of subfigures is confusing. Recommend switching the order of the year and sub-figure numbering. For example, use "(a) 1611; (b) 1810; ..." instead

AR:     We thank the Reviewer for this suggestion. Done.

**Additional references**

Carniello, L., Defina, A., & D'Alpaos, L. (2009). Morphological evolution of the Venice lagoon: Evidence from the past and trend for the future. *Journal of Geophysical Research: Earth Surface*, *114*(F4), F04002. https://doi.org/10.1029/2008JF001157

D'Agostino, R. B. (1986). *Goodness-of-Fit Techniques*. (R. B. D'Agostino & M. A. Stephens, Eds.). Routledge. https://doi.org/10.1201/9780203753064

D'Alpaos, L. (2010a). *Fatti e misfatti di idraulica lagunare. La laguna di Venezia dalla diversione dei fiumi alle nuove opere delle bocche di porto. Istituto Veneto di Scienze, Lettere e Arti*. Venice: Istituto Veneto di Scienze, Lettere e Arti.

D'Alpaos, L. (2010b). *L'evoluzione morfologica della laguna di Venezia attraverso la lettura di alcune mappe storiche e delle sue mappe idrografiche*. Istituto Veneto di Scienze, Lettere e Arti.

Finotello, A., Tognin, D., Carniello, L., Ghinassi, M., Bertuzzo, E., & D'Alpaos, A. (2023). Hydrodynamic Feedbacks of Salt-Marsh Loss in the Shallow Microtidal Back-Barrier Lagoon of Venice (Italy). *Water Resources Research*, *59*(3). https://doi.org/10.1029/2022WR032881

Silvestri, S., D'Alpaos, A., Nordio, G., & Carniello, L. (2018). Anthropogenic Modifications Can Significantly Influence the Local Mean Sea Level and Affect the Survival of Salt Marshes in Shallow Tidal Systems. *Journal of Geophysical Research: Earth Surface*, *123*(5), 996–1012. https://doi.org/10.1029/2017JF004503

Stephens, M. A. (1974). EDF Statistics for Goodness of Fit and Some Comparisons. *Journal of the American Statistical Association*, *69*(347), 730. https://doi.org/10.2307/2286009

Thode, H. C. (2002). *Testing For Normality* (Vol. 164). CRC Press. https://doi.org/10.1201/9780203910894

---

## Author Comment (AC2)

**Author Response to Reviews of Earth Surface Dynamics Manuscript egusphere-2023-319**

**Statistical characterization of erosion and sediment transport mechanics in shallow tidal environments. Part 1: erosion dynamics**

Andrea D'Alpaos[1], Davide Tognin[1,2], Laura Tommasini[1], Luigi D'Alpaos[2], Andrea Rinaldo[2,3], and Luca Carniello[2]

[1] *Department of Geosciences, University of Padova, Padova, Italy*
[2] *Department of Civil, Environmental, and Architectural Engineering, University of Padova, Padova, Italy*
[3] *Laboratory of Ecohydrology ECHO/IEE/ENAC, Ècole Polytechnique Fèdèrale de Lausanne, Lausanne, Switzerland*
Correspondence: Davide Tognin (davide.tognin@unipd.it)

**Summary**

The authors are grateful to the editorial board and the reviewers for their thoughtful and constructive comments on our paper, which significantly improved the manuscript and how our findings are communicated.
Following the Reviewers' suggestions, we have carefully revised the introduction in order to better highlight how the proposed approach aims to contribute to filling the knowledge gap in long-term morphodynamic modelling and to better frame the potential applicability of this approach.
Moreover, the revised manuscript now includes a more detailed description of the numerical hydrodynamic model used in our analysis and its calibration procedure.
Finally, as suggested by the Reviewers, we provided additional details about some modelling choices that were not properly justified in the previous version of the manuscript. In particular, now we extensively discuss the choice of boundary conditions and the threshold shear stress to apply the peak-over-threshold analysis.
Overall, in the new version of the manuscript, we consistently revised the main text and importantly expanded the Supplementary Information, by adding the detailed model description and figures S2 to S6.
In the following, we discuss in detail all Reviewers' comments and show how we addressed them, referencing line numbers in the revised version of the manuscript with the track changes.
Please note that the Reviewers' comments are in blue, our detailed responses are in black, and the text of the manuscript is framed.

*Legend*

RC:    Reviewer Comment

AR:    Author Response

☐ :    Modified manuscript text

*Note*: References to reviewers' comments are indicated as RCx.x and numbered progressively.

**Reply to Reviewer #2**

RC2.0: This is a very interesting paper combining a modeling approach and a statistical analysis of erosion dynamics in the Venice Lagoon. The paper is well written and the findings of potential interest for assessing long-term erosion processes in a computationally-efficient way. I only have minor remarks.

AR:     We thank the Reviewer for his/her positive comments on our manuscript and for his/her insightful suggestions that contributed to improving the quality and clarity of our manuscript. Please, find in the following the responses to each detailed comment.

RC2.1: I miss a discussion about the potential applicability of the marked Poisson model for intertidal flats in other environments (e.g., with a different wind regime and/or different tidal regime).

AR:     We agree with the Reviewer that a discussion about the applicability of this approach helps the reader to better understand potential and limitations.
        First of all, we revised the introduction to better frame the problem and highlight that the proposed approach aims to be used when stochastic processes, such as wind waves and storm surges, play a fundamental role in the morphological evolution of tidal systems. The revised text now reads:

[revised manuscript text omitted]

RC2.2: I also miss a discussion on the sensibility of the study results and conclusions to some key parameters (e.g., critical bottom shear for erosion, here fixed at 0.4 Pa, but greatly varying in the literature).

AR: We understand the Reviewer's point of view and we recognize that, in the first version of the manuscript, the choice of the critical shear stress and its implications were too condensed and, therefore, would benefit from a more detailed comment.
We added a more detailed explanation in the Method section as follows:

(line 232) In this work, at any location within each considered configuration of the Venice Lagoon, we used the peak-over-threshold (POT) theory (Balkema and de Haan, 1974) to analyze the temporal and spatial evolution of the total BSS, $\tau_{wc}$.
*In general, the selection of the threshold for the POT method must satisfy two contrasting requirements. On the one hand, the threshold must be large enough to discern stochastic events from the deterministic background. On the other hand, the threshold should not be too high to avoid the loss of important information and the need for a much longer time series to compute meaningful statistics, because of the lower number of threshold exceedances. Moreover, the extreme value theory postulates the general emergence of Poisson processes whenever the censoring threshold is high enough (Cramér and Leadbetter, 1967). To comply with these requirements, in the present analysis, the threshold is maintained well below the*

*maximum observed values, in order to remove only the background modulation induced by tidal currents without losing significant information on the stochastic wave-driven erosion process.*

*In applying the POT method to BSS time series, setting the threshold equal to a critical BSS for erosion, $\tau_c$, presents the non-trivial advantage of preserving also the physical meaning of the erosion mechanism. Values of critical BSS for erosion for fine, cohesive mixtures typical of shallow tidal settings largely vary in the literature and are affected by multiple physical and biotic factors (Mehta et al., 1989). Erosion shear stress from in-situ measurements on the tidal flats of the Venice Lagoon ranges between 0.2 and 2.3 Pa (0.7 ± 0.5 Pa - median ± standard deviation), with values higher than 0.9 Pa usually recorded within densely vegetated patches (Amos et al., 2004). In the present analysis, we cannot take into account the role of the biotic component, because of the impossibility to reconstruct the spatial distribution of vegetated tidal flats in the ancient configurations of the Venice Lagoon. For all the above reasons and following the approach suggested by D'Alpaos et al. (2013), we set the critical shear stress, $\tau_c$, equal to 0.4 Pa for all the selected historical configurations of the Venice Lagoon.*

*Before performing the POT analysis, the time series of BSSs were processed by applying a moving average filter, in order to remove spurious upcrossings and downcrossings of the prescribed threshold. This low-pass filter with a time window of 6 hours removes short-term fluctuations, preserving the modulation given by the semidiurnal tidal oscillation. Thanks to this preprocessing procedure, over-threshold events satisfy the independence assumption required by the statistical analysis applied.*

Concerning the sensitivity analysis, we added a paragraph as follows:

*(line 281) The result of modelling erosion events as a Poisson process stands regardless of the specific value of the censoring threshold selected for the POT analysis, provided that it is high enough to exclude deterministic exceedances, and this is confirmed also by the sensitivity analysis performed by D'Alpaos et al. (2013) on the present-day configuration of the Venice Lagoon. Indeed, when considering too low values of the threshold (e.g., $\tau_c = 0.2$ Pa), deterministic exceedances driven by tidal currents occur and make the interarrival time not exponentially distributed. On the contrary, as the threshold value increases (e.g., $\tau_c \geq 0.6$ Pa), the KS test is still verified, thus confirming that the process remains Poisson for increasing censoring thresholds (see Figure 6 in D'Alpaos et al. (2013) for further details).*

For the Reviewer's convenience, we report here Figure 6 from D'Alpaos et al. (2013) showing the results of the KS test using different BSS thresholds.

[Figure]

Figure 1. Spatial distribution of Kolmogorov-Smirnov (KS) test at a significance level (α = 0.05) on interarrival times assuming a threshold value for the shear stress equal to: $\tau_c = 0.2$ Pa (upper panel), $\tau_c = 0.4$ Pa (central panel) and $\tau_c = 0.6$ Pa (lower panel) (from D'Alpaos et al., 2013).

RC2.3: Line 67: Not clear if this tidal range or amplitude.

AR:     We thank the Reviewer for noting it. We amended the text as follows:

> (line 118) In the present-day morphology, the lagoon is connected to the sea with three inlets, namely Lido, Malamocco, and Chioggia (Figure 1), through which *the tide propagates within the back-barrier system. The tidal regime is semidiurnal with a maximum tidal amplitude of about 0.75 m, typical of the northern Adriatic Sea*  (D'Alpaos et al., 2013, Valle-Levinson et al., 2021).

RC2.4: Line 105: Provide the Strickler equation.

AR:     We thank the Reviewer for the suggestion. This piece of information is indeed crucial in this work and was actually missing in the original version of the manuscript. We provided the formulation adopted in the model of the Strickler equation by modifying the text as follows:

> (line 165) The bottom shear stress induced by currents, $\tau_{tc}$, is evaluated using the Strickler equation considering the case of a turbulent flow over a rough wall, *which can be written as (Defina, 2000)*

$$\tau_{tc} = \rho \, g \, Y \left( \frac{|q|}{K_s^2 H^{10/3}} \right) q$$

*where ρ is water density, g is the gravity acceleration, Y is the effective water depth, defined as the volume of water per unit area actually ponding the bottom, q is the flow rate per unit width, $K_s$ is the Strickler roughness coefficient, and H is an equivalent water depth that accounts for the typical height of ground irregularities.*

**RC2.5: Line 112: Provide the BSS induced by wind waves.**

AR:   To complete the description of the formulation of BSS adopted in the model, we modified the text as follows:

(line 176) The wind-wave module computes the bottom shear stress induced by wind waves as (*Carniello, 2005*)

$$\tau_{ww} = \frac{1}{2} \rho \, f_w \, u_m^2$$

*where $u_m$ is the maximum horizontal orbital velocity associated with wave propagation and $f_w$ is the wave friction factor. According to the linear theory, the bottom velocity $u_m$ can be evaluated as*

$$u_m = \frac{\pi H_w}{T \, \sinh(kh)}$$

*where $H_w$ is the wave height, T denotes the wave period, k is the wave number, and h is the water depth. The wave friction factor can be approximated as (Soulsby, 1997)*

$$f_w = 1.39 \left[ \frac{u_m T}{2 \, \pi (D_{50}/12)} \right]^{-0.52}$$

*where $D_{50}$ is the median grain size.*

**RC2.6: Lines 144-145: I am wondering if the largest exceedance of the threshold is really the most appropriate here. I feel that the integral of the exceedance makes more sense, as it will determine the total amount of sediments that will be eroded during that event. Can you comment on that?**

AR:   We totally agree with the Reviewer that the integral of the exceedance is the best metric to describe the total amount of eroded sediment during an event. Indeed, we computed the erosion work, which exactly matches this definition and fits this purpose (see Eq. 2 and 3 in the first version of the manuscript). However, the description of the process solely through the integral does not allow to understand whether the variation in erosion depends on an intensity or on a duration variation. Instead, describing the processes using these two variables (together with the interarrival time to provide the frequency), besides being simple and intuitive, does not prevent the computation of more specific metrics, such as the over-threshold integral (i.e., erosion work) which can be approximated by their combination (see Table 1 and Supplementary Figure S16).
To better justify this choice, we added a comment in the revised manuscript as follows:

(line 256) The POT method allowed us to identify:
- the interarrival time of over-threshold events, defined as the time between two consecutive upcrossings of the threshold;
- the duration of over-threshold events, that is the time elapsed between any upcrossing and the subsequent downcrossing of the threshold;
- its intensity, calculated as the largest exceedance of the threshold in the time elapsed between an upcrossing and the following downcrossing.

> *These three random variables synthetically characterize the over-threshold erosion events and can be combined to obtain further metrics to describe the erosion process (e.g., see the erosion work defined later on).*

 This paragraph seems central to the entire paper. However, it is very short and does not cite any reference where the supporting theory is fully developed. I expect more details to support the theory, either as supplementary material or as cited literature.

AR:     We probably took for granted that reader is familiar with stochastic processes, but we agree with the Reviewer that it is better to provide some additional information for the interested reader. We added the reference to these two classic textbooks which provide both a general overview of stochastic processes and detailed explanations of the properties of the Poisson process:

- Cramér, H. and Leadbetter, M. R.: Stationary and related stochastic processes, John Wiley & Sons, Ltd, New York, 1967.
- Gallager, R. G.: Stochastic Processes: Theory for Applications, Cambridge University Press, https://doi.org/10.1017/CBO9781139626514, 2013

RC2.8: Lines 167-171: This should be in the methods section.

AR:     We agree with the Reviewer. As reported in our response to RC2.2, we moved this paragraph to the method section (line 137 of the first version of the manuscript).

RC2.9: Lines 184-186: Results in Figures 4-6 correspond to areas in red or yellow in Figure 3. Why not to areas in red only? Can you elaborate on that choice? Is it not relevant whether intensity and/or duration are exponentially distributed random variables?

AR:     We agree that the choice of showing mean values of intensity and durations also where they are not exponentially distributed needs an additional comment to be better understood.
        Whether intensity and duration can be described by exponential distributions does not affect the chance to model erosion as a Poisson process, which indeed relies only on the exponential distribution of interarrival times, but it can simplify the setup of the final stochastic framework. However, even if these marks of the Poisson process cannot be described by an exponential distribution, mean values of peak excess and duration can still be considered informative of the trend of these random variables and, thus, it is still worth showing them in the figures.
        We reported this justification in the revised text, which now reads:

> (line 301) *We analyzed the time series of computed total BSSs, $\tau_{wc}$, at any element of the computational grids reproducing the six selected configurations of the Venice Lagoon on the basis of the POT method, in order to characterize the over-threshold erosion events in terms of interarrival times, peak excess and duration. The KS test is then performed in each element of the six domains in order to test where interarrival times can be described by an exponential distribution and thus, the over-threshold erosion events can be modelled as a Poisson process. We performed the KS test also on peak excess and duration to test if these marks of the process can also be described by exponential distributions. Whether peak excess and duration can be described by exponential distributions does not affect the chance to model erosion as a Poisson process, which indeed relies only on the exponentiality of interarrival times, but it can simplify the setup of the final stochastic framework.*
> *Therefore, in the spatial distribution of KS test results (Figure 3),*  *we distinguished:*

- the dark blue area, where the KS test is not verified for the interarrival time, i.e. wave-induced erosion events can not be described as a Poisson process;
- the red area, where the KS test is verified for all the three considered stochastic variables, namely interarrival times, intensity, and duration, i.e. wave-induced erosion events are indeed a marked Poisson process where its marks, intensity and duration, are *also* exponentially distributed random variables;
- the yellow area, where the KS test is verified for the interarrival time but it is not verified for the intensity and/or duration, i.e. wave-induced erosion events are a marked Poisson process but at least one between intensity and duration is not an exponentially distributed random variable.

The mean interarrival times (Figure 4), mean peak excesses (Figure 5) and mean durations of over-threshold erosion events (Figure 6) in the six selected configurations of the Venice Lagoon are shown in every location where the KS test is satisfied for interarrival times (Figure 3), and, thus, erosion events, can be described as a Poisson process. *Mean peak excess and mean duration are shown also where at least interarrival times are exponentially distributed (i.e., yellow areas in Figure 3) because mean values are anyway considered to be informative and erosion events can still be modelled as a Poisson process, although the marks are described by a distribution different from the exponential one.*

RC2.10: Lines 233-236: Please elaborate on the morphological features that remain the same through the last four centuries.

AR:   To complete the description of the lagoon morphology, we added a comment in the geomorphological setting section, that now reads:

(line 140) *Only in the northern lagoon, the morphological degradation was less pronounced because of the sheltering effect provided by the mainland against the north-easterly Bora wind and the less intense human pressure. Therefore, the northern basin displays also in the present-day configuration relatively shallow intertidal flats and larger salt-marsh areas, compared to the central and southern lagoon (Figure 2f)*

Moreover, we added a comment to highlight the effects on interarrival times of preserved morphological features:

(line 374) *On a tidal flat in the northern lagoon named ``Palude Maggiore'' (see station S1 in Figure 1a), as in most areas of the lagoon, the mean interarrival time $\bar{\lambda}_t$ between two subsequent over-threshold events increases through time (Figure 8a). This is because this area preserved the same morphological features, i.e. relatively shallow tidal flats protected by the mainland and salt marshes, over the last four centuries.*

**Additional references**

Carniello, L., Defina, A., & D'Alpaos, L. (2009). Morphological evolution of the Venice lagoon: Evidence from the past and trend for the future. *Journal of Geophysical Research: Earth Surface*, *114*(F4), F04002. https://doi.org/10.1029/2008JF001157

D'Agostino, R. B. (1986). *Goodness-of-Fit Techniques*. (R. B. D'Agostino & M. A. Stephens, Eds.). Routledge. https://doi.org/10.1201/9780203753064

D'Alpaos, L. (2010a). *Fatti e misfatti di idraulica lagunare. La laguna di Venezia dalla diversione dei fiumi alle nuove opere delle bocche di porto. Istituto Veneto di Scienze, Lettere e Arti*. Venice: Istituto Veneto di Scienze, Lettere e Arti.

D'Alpaos, L. (2010b). *L'evoluzione morfologica della laguna di Venezia attraverso la lettura di alcune mappe storiche e delle sue mappe idrografiche*. Istituto Veneto di Scienze, Lettere e Arti.

Finotello, A., Tognin, D., Carniello, L., Ghinassi, M., Bertuzzo, E., & D'Alpaos, A. (2023). Hydrodynamic Feedbacks of Salt-Marsh Loss in the Shallow Microtidal Back-Barrier Lagoon of Venice (Italy). *Water Resources Research*, *59*(3). https://doi.org/10.1029/2022WR032881

Silvestri, S., D'Alpaos, A., Nordio, G., & Carniello, L. (2018). Anthropogenic Modifications Can Significantly Influence the Local Mean Sea Level and Affect the Survival of Salt Marshes in Shallow Tidal Systems. *Journal of Geophysical Research: Earth Surface*, *123*(5), 996–1012. https://doi.org/10.1029/2017JF004503

Stephens, M. A. (1974). EDF Statistics for Goodness of Fit and Some Comparisons. *Journal of the American Statistical Association*, *69*(347), 730. https://doi.org/10.2307/2286009

Thode, H. C. (2002). *Testing For Normality* (Vol. 164). CRC Press. https://doi.org/10.1201/9780203910894

---

## Author Response (AR2)

**Author Response to Reviews of Earth Surface Dynamics Manuscript egusphere-2023-319**

**Statistical characterization of erosion and sediment transport mechanics in shallow tidal environments. Part 1: erosion dynamics**

Andrea D'Alpaos[1], Davide Tognin[1,2], Laura Tommasini[1], Luigi D'Alpaos[2], Andrea Rinaldo[2,3], and Luca Carniello[2]

[1] *Department of Geosciences, University of Padova, Padova, Italy*
[2] *Department of Civil, Environmental, and Architectural Engineering, University of Padova, Padova, Italy*
[3] *Laboratory of Ecohydrology ECHO/IEE/ENAC, Ècole Polytechnique Fèdèrale de Lausanne, Lausanne, Switzerland*
Correspondence: Davide Tognin (davide.tognin@unipd.it)

**Summary**

The authors wish to thank the Editorial Board and the Reviewers for their suggestions. We carefully considered and extensively discussed the possibility of merging the two papers. However, we hold major reservations about merging the two contributions, as we firmly believe that our work can be most effectively conveyed through two separate papers.

As explained more in detail in the following, the main rationale for keeping the two manuscripts separate is content-related, as each paper conveys a distinct message. The overarching contribution of the two companion papers is to test the hypothesis of using random processes to upscale morphodynamic models. However, this cannot be limited to the analysis of erosion events presented in Part 1, because suspended sediment dynamics is not solely influenced by local resuspension but also by advective and mixing processes occurring at the basin scale. Therefore, the characterization of both erosion events and suspended sediment dynamics as Poisson processes is necessary to test the possibility of implementing a synthetic modelling framework accounting for erosion and deposition. This highlights that the two papers are not mere repetitions but rather they address complementary questions on different morphological processes.

To better highlight the complementarity of these works, we have deeply revised the introduction of both papers, as detailed below.

*Legend*

RC:    Reviewer Comment

AR:    Author Response

☐ :    Modified manuscript text

*Note*: References to reviewers' comments are indicated as RCx.x and numbered progressively.

AR: The main rationale for maintaining the two manuscripts separated is content-related, as each paper has its own message. The most significant contribution of our study is to test the hypothesis to use random processes to upscale morphodynamics models. When describing morphodynamic changes, both erosive and depositional processes play a fundamental role. Erosion is generally related to the local BSS and deposition to the available SSC. The peak-over-threshold analysis of BSS presented in Part 1 proves that erosion dynamics can be modelled as a Poisson process. However, this offers only a partial perspective, as it does not address the possibility of modelling depositional dynamics as a stochastic process. Indeed, SSC is not solely influenced by local erosion because of advective and dispersive processes occurring at the basin scale, and, hence, must be analyzed independently. Therefore, the novelty of Part 2 lies in demonstrating that spatio-temporal dynamics of SSC can also be modelled as a random process, which is not proved in Part 1.

The characterization of both BSS and SSC as Poisson processes is necessary to test the possibility of implementing a synthetic modelling framework accounting for erosion and deposition. This highlights the difference and the complementarity of the results and clearly demonstrates that Part 2 is not a mere repetition of Part 1 but rather a fundamental component of our research.

To further substantiate this concept, we modified the introduction of Part 1 as follows:
* * *
Manuscript egusphere-2023-319

(line 60) *A different perspective would be to directly consider the stochasticity of morphodynamic processes. From this point of view, the first step is to test the possibility of setting up a statistically-based framework in order to generate synthetic, yet reliable, time series to model the morphodynamic evolution on long-term time scales and compare possible scenarios in a computationally-effective way through the use of independent Monte Carlo realizations. Although the statistical characterization of the long-term behaviour of several geophysical processes is becoming increasingly popular in hydrology and geomorphology (e.g., Rodriguez-Iturbe et al., 1987; D'Odorico and Fagherazzi, 2003; Botter et al., 2007; Park et al., 2014), applications to tidal landscapes are still quite rare (D'Alpaos et al., 2013; Carniello et al., 2016).*

*The morphological evolution of tidal systems can be described by Exner's equation:*

$$(1-n)\frac{\partial z_b}{\partial t} + \nabla \boldsymbol{q_b} = D - E \qquad (1)$$

*where $n$ is the bed porosity, $z_b$ is the bed elevation, $\boldsymbol{q_b}$ is the bedload, $D$ and $E$ are the deposition and entrainment rates of sediment, respectively. In mud-dominated tidal systems, sediment is primarily transported in suspension and the bedload is negligible, hence, the bed level changes can be determined by accurately describing erosion and deposition. Erosion, E, is directly influenced by the local bottom shear stress (BSS), which results from the interaction between tidal currents and wind waves in shallow tidal systems (Green and Coco, 2014). Instead, deposition, D, is linked to the suspended sediment concentration (SSC). However, SSC is largely affected by advection and dispersion processes at a larger scale and, therefore cannot be solely determined by local resuspension. Consequently, to effectively model bed-level variations, it is essential to accurately describe both BSS and SSC. This contribution focuses on characterizing BSS, while the analysis of SSC is presented in the companion paper (Tognin et al., 2023).*
* * *
In the introduction of Part 2, we added a very brief recall to Exner's equation presented in Part 1 and discussed the differences in the analysis of SSC as follows:

Manuscript egusphere-2023-320

(line 51) *A comprehensive understanding of morphological processes is key to addressing management and restoration strategies for shallow tidal landscapes. The morphodynamic evolution of these systems can be described by Exner's equation:*

$$(1 - n)\frac{\partial z_b}{\partial t} + \nabla \boldsymbol{q_b} = D - E \qquad (1)$$

*where $n$ is the bed porosity, $z_b$ is the bed elevation, $\boldsymbol{q_b}$ is the bedload, D and E are the deposition and entrainment rates of sediment, respectively. Bedload is usually negligible in mud-dominated tidal systems, because sediment transport mainly occurs in suspension, and, thus, the bed level changes are essentially a function of erosion and deposition processes. In order to complete the stochastic framework introduced by D'Alpaos et al. (2023) for the description of erosion events, this study deals with the statistical characterization of suspended sediment concentration (SSC), considered a proxy for depositional processes.*
*Suspended sediment dynamics in shallow tidal systems are influenced by different hydrodynamic and sedimentological factors that vary over a wide range of spatial and temporal scales (Woodroffe, 2002; Masselink et al., 2014). Both tide and waves represent key drivers controlling sediment entrainment and transport in shallow tidal environments (Wang, 2012), with stochastic wave-forced resuspension occasionally increasing by far cyclic tide-driven sediment reworking, especially under storm conditions. Wave resuspension together with tide- and wave-driven sediment transport give rise to advection and dispersion mechanisms leading to basin-wide sediment movement, which largely affect local suspended sediment dynamics (e.g., Nichols and Boon, 1994; Carniello et al., 2011; Green and Coco, 2014). Owing to the complexity of the underlying processes, suspended sediment dynamics in shallow tidal systems is rather entangled and it is not only linked to the local bottom resuspension. Therefore, to effectively describe suspended sediment transport in shallow tidal systems, a dedicated analysis is required.*
*Several numerical models have been developed to describe sediment transport and different techniques have been proposed to upscale the effects on the morphological evolution of tidal systems. For instance, explorative point-based models are extensively used to understand the relative importance of sediment transport processes, because of their simplified parametrization as well as their great conceptual value (Murray, 2007). Furthermore, their reduced computational burden is ideal for investigating trends over long-term time scales. For these reasons, point-based models have been largely adopted, for example, to examine salt-marsh fate under different sea level rise scenarios at the century time scale (D'Alpaos et al., 2011; Fagherazzi et al., 2012). However, point-based models potentially miss spatial dynamics associated with sediment transport and, hence, might fail to represent interactions between different morphological units. More detailed, process-based models can fill this gap and account for sediment fluxes between different points up to the whole basin scale (e.g. Lesser et al., 2004; Carniello et al., 2012). But, because of the explicit description of the short-term interaction between hydrodynamics and sediment transport, the application of process-based models to the long-term time scale is often computationally expensive or even prohibitive.*